# Seismic evidence for uniform crustal accretion along slow-spreading ridges in the equatorial Atlantic Ocean

Zhikai Wang [1] & Satish C. Singh [1]

The crustal accretion along mid-ocean ridges is known to be spreading-rate dependent. Along fast-spreading ridges, two-dimensional sheet-like mantle upwelling creates relatively uniform crust. In contrast, the crust formed along slow-spreading ridges shows large along-axis thickness variations with thicker crust at segment centres, which is hypothesised to be due a three-dimensional plume-like mantle upwelling or due to focused melt migration to segment centres. Using wide-angle seismic data acquired from the equatorial Atlantic Ocean, here we show that the crustal thickness is nearly uniform (~5.5 km) across five crustal segments for crust formed at the slow-spreading Mid-Atlantic Ridge with age varying from 8 to 70 Ma. The crustal velocities indicate that this crust is predominantly of magmatic origin. We suggest that this uniform magmatic crustal accretion is due to a two-dimensional sheet-like mantle upwelling facilitated by the long-offset transform faults in the equatorial Atlantic region and the presence of a high concentration of volatiles in the primitive melt in the mantle.

The oceanic crust covers ~60% of the Earth's surface and is continuously generated along the ~65,000 km-long divergent plate boundaries at the Mid-Oceanic Ridges (MORs)[1]. The MORs are partitioned into tens to hundreds kilometre-long first-order segments by oceanic transform faults (TFs)[1]. Between two oceanic TFs, these ridge segments are further divided into smaller ridge segments (second-order segments) by non-transform offsets (NTOs) at slow-spreading ridges and overlapping spreading centres (OSCs) at fast-spreading ridges[1].

The crustal accretion along MORs and the resulting along-axis crustal thickness variation are known to be spreading-rate dependent. At the fast-spreading ridges, such as the East-Pacific Rise (EPR), the oceanic crust exhibits relatively uniform thickness without or with only a modest crustal thinning (≤1.6 km; yellow triangles in Fig. 1) at oceanic TFs and OSCs[2–6], interpreted to result from a two-dimensional (2-D) sheet-like mantle upwelling beneath these ridges[3] and/or rapid ductile flow of hot lower crust along these ridges[7]. In contrast, the studies using gravity data[3,8–11] collected along the slow-spreading Mid-Atlantic Ridge (MAR) reveal systematically significant reduction (up to 50%) in crustal thickness from segment centres to the associated distal ends at

TFs and NTOs. Crustal thickness measured using active-source seismic data also shows large and systematic along-axis crustal thickness variations at the MAR, where the crust at distal ends could be up to ~3.2–5.7 km thinner than that at the segment centres[9,12–18] (yellow dots in Fig. 1). All these observations suggest a focused magmatic crustal accretion along the slow-spreading ridges, which is interpreted to result from either a three-dimensional (3-D) plume-like mantle upwelling[3,8] or from a 3-D melt migration to the segment centres at the base of the lithosphere beneath slow-spreading ridges[13,19,20]. However, these previous studies on the MAR are generally concentrated within or close to the spreading centres, sampling only young (≤2 Ma) oceanic crust. In contrast, old oceanic crust formed along the MAR shows different patterns of segment-scale crustal thickness variation. For example, although large segment-scale crustal thickness variations are observed for ~5 and ~65 Ma old crust in the North Atlantic Ocean[17,21,22], the crustal thinning is only on the scale of ~2.8–3.4 km (blue dots in Fig. 1), smaller than that observed along or near the ridge axis. Furthermore, the ~130–150 Ma old crust in the North Atlantic Ocean shows very little variations in crustal thickness away from fracture zones (FZs), the fossil traces of the oceanic TFs, but a thin crust (2–4 km) at

[1]Université Paris Cité, Institut de Physique du Globe de Paris, CNRS, 1 rue Jussieu, Paris 75238, France. e-mail: zwang@ipgp.fr

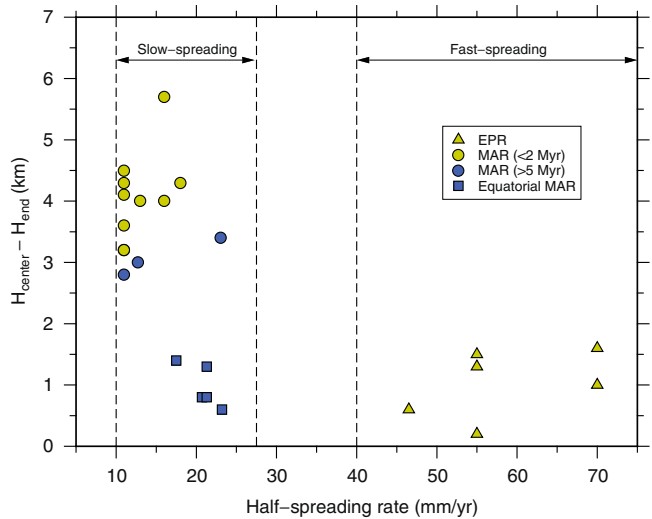

**Fig. 1 | Crustal thickness variation versus spreading rate.** Crustal thickness (H) difference between segment centres and segment ends as a function of spreading rate. Only the crustal thickness constrained by active-source seismic data is considered. The crustal thickness data from the Atlantic Ocean are selected following two criteria: (1) systematically along-axis crustal thinning is observed within the second-order ridge segment and (2) the crustal thicknesses at segment centre and at least one segment end are measured. The blue squares show the difference between the average crustal thickness at the transform fault, fracture zone and pseudo-fault region and the average crustal thickness of segments obtained in this study. When the crustal thicknesses of both ends of a segment are available, the one with thinner crust is plotted. The crust at the ends of slow-spreading ridges is generally ≥2.8 km thinner than that at the associated segment centres. Data for East Pacific Rise (EPR), South and North Mid-Atlantic Ridge (MAR) are given in Supplementary Table 2. The thin dashed vertical lines mark the boundaries of fast- and slow-spreading ridges[79].

the FZs[23,24] in a 10–20 km wide zone. Moreover, a recent study from the South Atlantic Ocean over 6.6–61.2 Ma old crust at 31°S also shows very small variations in crustal thickness over 40–60 km lateral distance[25], indicating that the 3-D mantle upwelling or 3-D melt migration hypotheses may not be valid everywhere in slow-spreading environments.

Plate separation at the slow-spreading ridges is accommodated by two contrasting modes of accretion[26]. For the regions where the melt supply is robust and sufficient, the magmatic accretion creates thick igneous crust[1,26]. In contrast, for the regions where melt supply is limited, plate separation is mainly accommodated through tectonic extension, which could emplace lower crustal and upper mantle rocks to the seafloor forming oceanic core complexes (OCCs) leading to thin crust[27].

Crustal P-wave velocity (Vp) obtained using wide-angle seismic data can be used to discriminate between these two different modes of crustal accretion. For example, magmatically accreted crust can be divided into two layers based on their velocities[28,29]. The upper crust (Layer 2) is characterised by a relatively low Vp (4.1–6.5 km/s) but a high vertical velocity gradient (~1–2 s⁻¹) whereas the lower crust (Layer 3) has a high Vp (6.5–7.1 km/s) but a significantly reduced velocity gradient (~0.1–0.2 s⁻¹). This seismic structure has been designated as the Penrose model which equates these layers to an upper crust composed of extrusive basalts and sheeted dikes overlying a gabbroic lower crust[1]. In contrast, the crustal Vp below the OCCs increases rapidly to >6.5–7.0 km/s within ~1.0–2.5 km below the seafloor[22,30–34], suggesting the presence of gabbroic rocks or partially serpentinised peridotite exhumed to shallow depths. However, any interpretation about the crustal structure based only on the Vp model is poorly constrained because of ambiguity in the velocity-lithology relationship[35]. As the Vp increases from ~4.5–5.0 km/s for a completely serpentinised peridotite to ~8.0 km/s for an unaltered peridotite[36,37], there is an overlap with the Vp (4.1–7.1 km/s) for magmatic oceanic crust[28]. On the other hand, laboratory measured

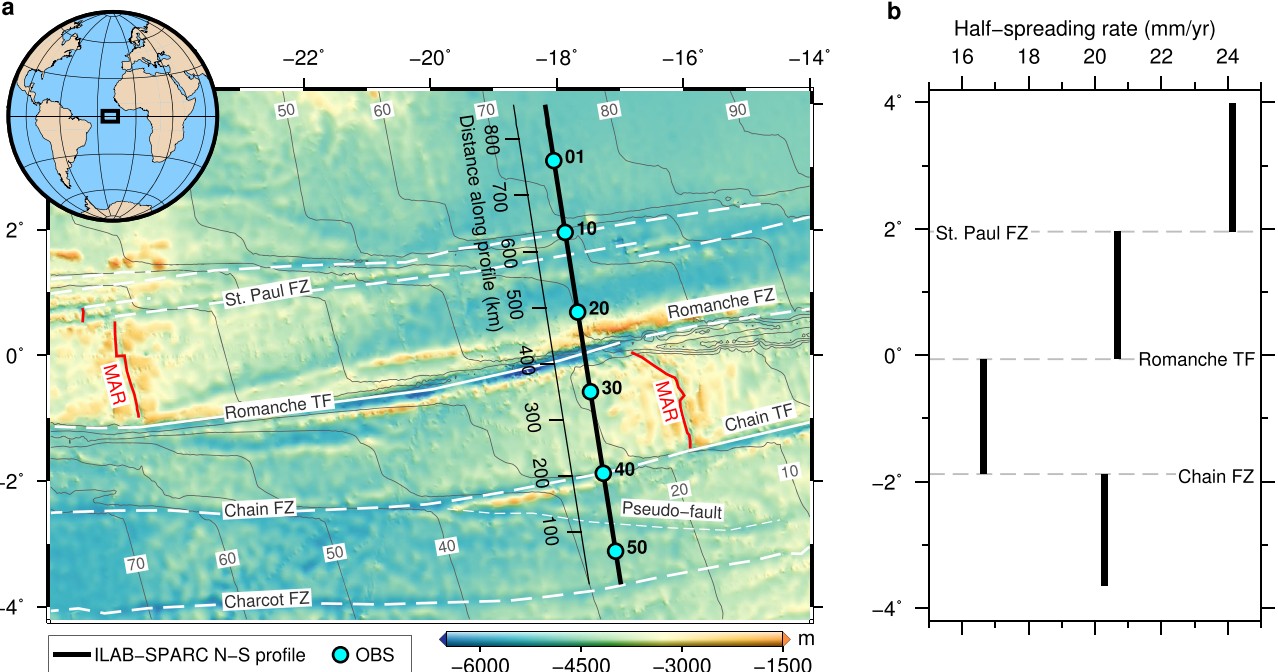

**Fig. 2 | Study area. a** Bathymetry map showing the Mid-Atlantic Ridge (MAR; red lines), transform faults (TFs; white solid lines) and fracture zones (FZs; white thick dashed lines) in the equatorial Atlantic Ocean. The seismic profile is shown as a black line, with the location of every tenth OBS marked by cyan dots. The age of the oceanic crust[42] is contoured and labelled every 10 Myr. The black rectangle in the globe inset shows the location of the study area. **b** Half-spreading rate[42] of the five segments at the time the measured crust were formed. As the half-spreading rate varied between 15 and 25 mm/yr, the equatorial MAR can be classified as a slow-spreading ridge (half-spreading rate between 10 and 27.5 mm/yr[79]).

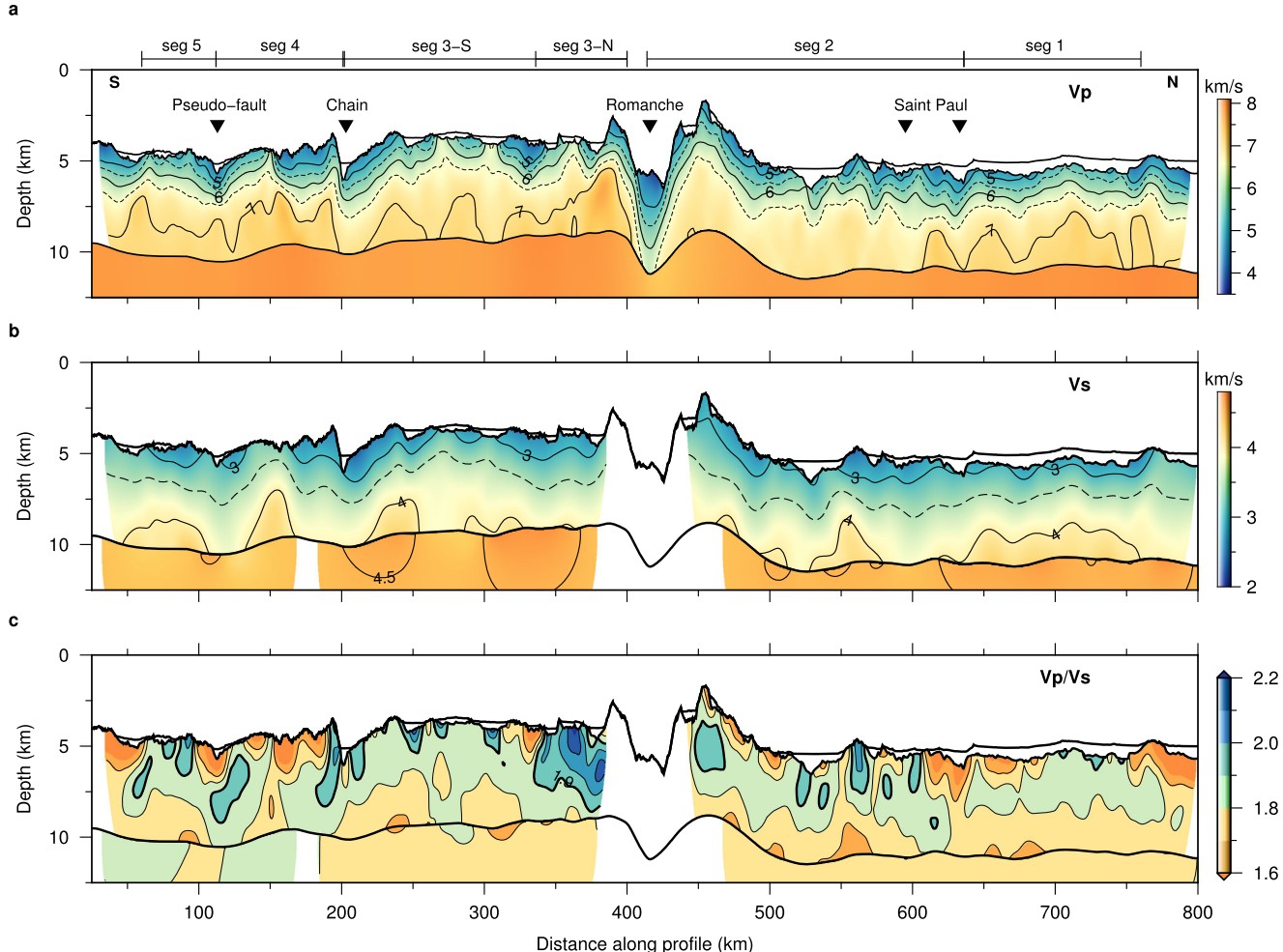

**Fig. 3 | Crustal and upper mantle velocity models.** Inverted P-wave velocity (Vp) model (**a**), S-wave velocity (Vs) model (**b**), and Vp/Vs ratio (**c**). The distance ranges of Segments 1 to 5 are marked at the top of **a**. The velocity colour scale for each figure is shown on the right. seg: Segment.

velocities of rocks from oceanic drill cores show that the magmatic crustal rocks (basalt and gabbro) generally have P and S-wave (Vs) velocity (Vp/Vs) ratios <1.9 at crustal pressure conditions[37,38] while the Vp/Vs ratio of serpentinised mantle rocks varies from ~1.78 to ~2.21 with increasing degree of alteration[36]. Therefore, the Vp/Vs ratio is a useful physical parameter for constraining crustal composition[39], hence the thickness of the crust and the crustal accretion process.

Our study area lies in the equatorial Atlantic Ocean covering five crustal segments formed at the slow-spreading MAR (Fig. 2a). The equatorial Atlantic Ocean started opening ~100–140 Ma ago[40,41]. The average half-spreading rate varied between 15 and 25 mm/yr in the past (Fig. 2b), and the current half-spreading rate is ~16 mm/yr[42]. In this region, three east-west striking mega-transform faults (the St. Paul, Romanche and Chain TFs) offset the MAR by a total of ~1800 km (Fig. 2a). The St. Paul TF system encompasses four TFs interrupted by three intra-transform ridge segments, generating a total offset of ~600 km[43]. Away from its eastern ridge-transform intersections (RTIs), the traces of the four TFs persist up to ~20 Ma old seafloor and then only two traces of the TFs can be identified on the older seafloor[44]. Our seismic profile transects the St. Paul FZs where the crustal age is ~70 Ma in the north and ~40 Ma in the south. The Romanche TF offsets the MAR by ~880 km[45]. The portion of the Romanche TF crossed by our seismic profile is characterised by a ~6-km-deep and ~40 km-wide valley with bounding walls shallowing to <3 km depth below sea level. The age contrast along our profile is ~32 Ma across the Romanche TF, with crustal ages of ~40 Ma in the north and ~8 Ma in the south. The seismic

profile crosses the Chain FZ at ~160 km west of the western Chain RTI, where the crustal age is ~10 Ma in the north and ~24 Ma in the south. Along our profile, the Chain FZ is characterised by a ~10-km-wide sedimented valley, which is bounded by a transverse ridge to the south and shows gradual seafloor shallowing to the north[46]. An oblique pseudo-fault is observed between the Chain and Charcot FZs, intersecting the Chain FZ at ~46 Ma to the west and a NTO within the ~17 Ma old lithosphere[46], showing a typical half V-shaped feature on the seafloor.

Here, we present results from an active-source seismic refraction experiment to estimate the Vp, Vs and Vp/Vs ratio. Based on these Vp, Vs and Vp/Vs models of the crust, we first determine the predominant mode of crustal formation for each crustal segment and then analyse the segment-scale crustal thickness variations to understand the mantle upwelling process along the slow-spreading MAR in the equatorial region.

## Results

The ~855-km-long seismic refraction profile was acquired in 2018 during the ILAB-SPARC experiment[43,45–47]. The seismic profile was deployed in an approximately N-S direction nearly parallel to the MAR, starting just north of the Charcot FZ in the south and extending to 70 Ma old crust ~230 km north of the St. Paul FZ (Fig. 2a). Fifty ocean bottom seismometers (OBSs) were deployed along the ~700 km part of the profile at an average spacing of 14.2 km. An airgun-array of 16 guns with a total volume of 81.77 litres was towed at 10 m depth below the

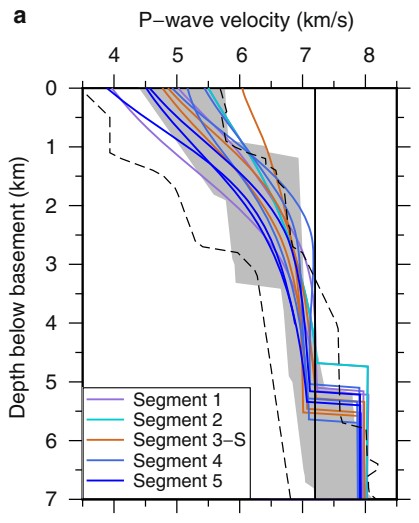
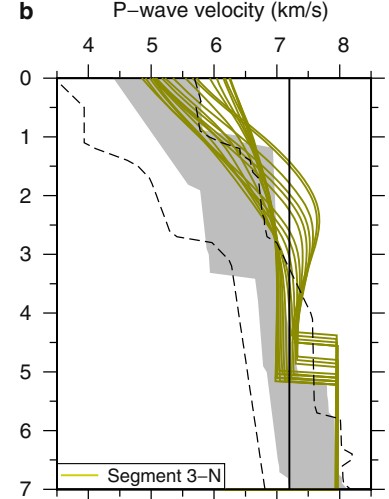

**Fig. 4 | One-dimensional P-wave velocity-depth profiles. a** Profiles extracted from Segments 1, 2, 3-S, 4, 5 (see the legends) with predominantly magmatic accreted crust. **b** Profiles extracted from Segment 3-N with predominantly tectonically formed crust. The black dashed lines represent the velocity envelope for the oceanic crust aged 59–150 Myr in the Atlantic Ocean[29]. The grey shading represents the velocity envelope for the slow-spreading oceanic crust aged >7.5 Myr[28]. The black solid lines show the 7.2 km/s velocity.

sea level and was fired at 300 m interval. Both P and S-wave arrivals were hand-picked on the pressure data recorded by hydrophones (Supplementary Figs. 1 and 2). Detailed analyses of these data demonstrate that the picked S-wave arrivals travel as P-waves in the water and sediments (if present), convert to S-wave at the igneous crust interface and travel as S-wave in the crust (See Methods; Supplementary Figs. 3–5). All the arrival picks were inverted using a 2-D ray-based travel time tomographic method[2,48] (See Methods; Supplementary Figs. 6–12) to obtain Vp and Vs models (Fig. 3a, b), which were used to compute Vp/Vs ratios (Fig. 3c), allowing to shed light on crustal accretion and mantle upwelling in the equatorial Atlantic region.

**Magmatically accreted crust versus tectonically controlled crust**
To determine the dominant mode of crustal accretion, we use a Vp/Vs ratio of 1.9 to discriminate between magmatic crust (Vp/Vs < 1.9) and serpentinised peridotite (Vp/Vs > 1.9) at crustal depth[39]. The tomographic results show two different types of crustal Vp structures for the five crustal segments characterised by distinct crustal Vp/Vs ratios (Fig. 3), which can be interpreted as resulting from two different modes of crustal accretion: magmatic accretion and tectonically controlled accretion.

The oceanic crust within Segments 1, 2, 3-S, 4 and 5 exhibits a two-layered Vp structure (Figs. 3a and 4a) characterised by distinct vertical Vp gradients. Using a vertical Vp gradient of 0.5 s⁻¹ to define the boundary between the upper and lower crust[49], the upper crust is ~1.9–2.3 km thick showing high vertical Vp gradients of ~0.66–0.80 s⁻¹ (Supplementary Table 1). Beneath the upper crust, the Vp in the lower crust, which is ~3.1–3.5 km thick, increases at a much-reduced velocity gradient of ~0.13–0.17 s⁻¹ (Supplementary Table 1). The maximum crustal Vp within these segments is generally <7.2 km/s (Figs. 3a and 4a). These crustal Vp structures fall in the velocity range typical for older slow-spreading magmatic oceanic crust[28,29] (Fig. 4a). These crustal segments are generally characterised by crustal Vp/Vs ratios less than 1.9 (Fig. 3c), indicative of basaltic and gabbroic rocks in the crust[39]. The Mohorovičić boundary (Moho), the crust/mantle boundary, is well constrained using wide-angle reflections (PmP) from the Moho. A rapid increase from crustal velocities to mantle velocities across the Moho is observed throughout these crustal segments, where the Vp increases abruptly by ~0.7–0.9 km/s across the Moho (Figs. 3a and 4a) to 7.8–8.1 km/s, the velocity corresponding to mantle peridotite. Both the Vp and Vp/Vs ratio values indicate that the

observed two-layered crustal velocity structure can be explained by the Penrose model[1] suggesting a predominantly magmatic accretion of crust throughout these crustal segments.

In contrast, the Segment 3-N between 340 and 400 km profile distance immediately south of the Romanche TF shows a rapid increase in the crustal Vp to ~7.7 km/s at ~2.2 km sub-basement depth (Figs. 3a and 4b). This anomalously high Vp exceeds the typical velocity for the normal oceanic crust (<7.2 km/s[28,29]). Conversely, the upper crustal Vs within Segment 3-N shows a similar velocity-depth variation as that within Segment 3-S. The crustal Vp/Vs ratios are larger than 1.9 in the upper crust (Fig. 3c), suggesting the presence of serpentinised peridotite, indicative of predominantly tectonic extension[39]. The rapid increase in the crustal Vp with depth can be interpreted as a gradual decrease in mantle serpentinisation. Assuming a Vp of 8.0 km/s and 5.0 km/s for 0% and 100% serpentinised peridotite[36,37], respectively, the observed ~7.7 km/s velocities represent ~10% mantle serpentinisation at ~2.2 km sub-basement depth. A recent petrological study of dredged highly deformed peridotites from the eastern Romanche RTI region identified fragments of old OCCs on the south flank of the Romanche TF[50]. This is consistent with our interpretation that immediately south of the Romanche TF, the plate separation is mainly accommodated by tectonically controlled accretion.

**Intra-segment crustal thickness variation within magmatic segments**
The thickness of the igneous crust is defined as the thickness between the basement and the seismically constrained Moho. Based on the crustal Vp and Vp/Vs structures, Segments 1, 2, 3-S, 4 and 5 (Fig. 3) are interpreted as consisting of magmatic crust. The average crustal thicknesses within these magmatic segments are ~5.4–5.6 km (Fig. 5), consistent with some previous estimates in this region[43,45,46]. The average crustal thicknesses of the five segments are ~500–700 m thinner than the global average of the old (>7.5 Ma) slow-spreading crust but fall within one standard deviation (6.1±1.0 km[28]). The oceanic crust shows a slight thinning in a narrow zone in the vicinity of the FZ, TF and pseudo-fault regions, where the thinnest crust at these tectonic discontinuities is <1.5 km thinner relative to the average of each segment (Fig. 5 and blue squares in Fig. 1). At the southern end of Segment 1, the crust thins by ~600 m to ~4.8 km thickness at the centre of the northern St. Paul FZ valley over <5 km distance (Fig. 5a, b). The crust within the ~20-km-wide Romanche transform valley is ~5.1 km thick on

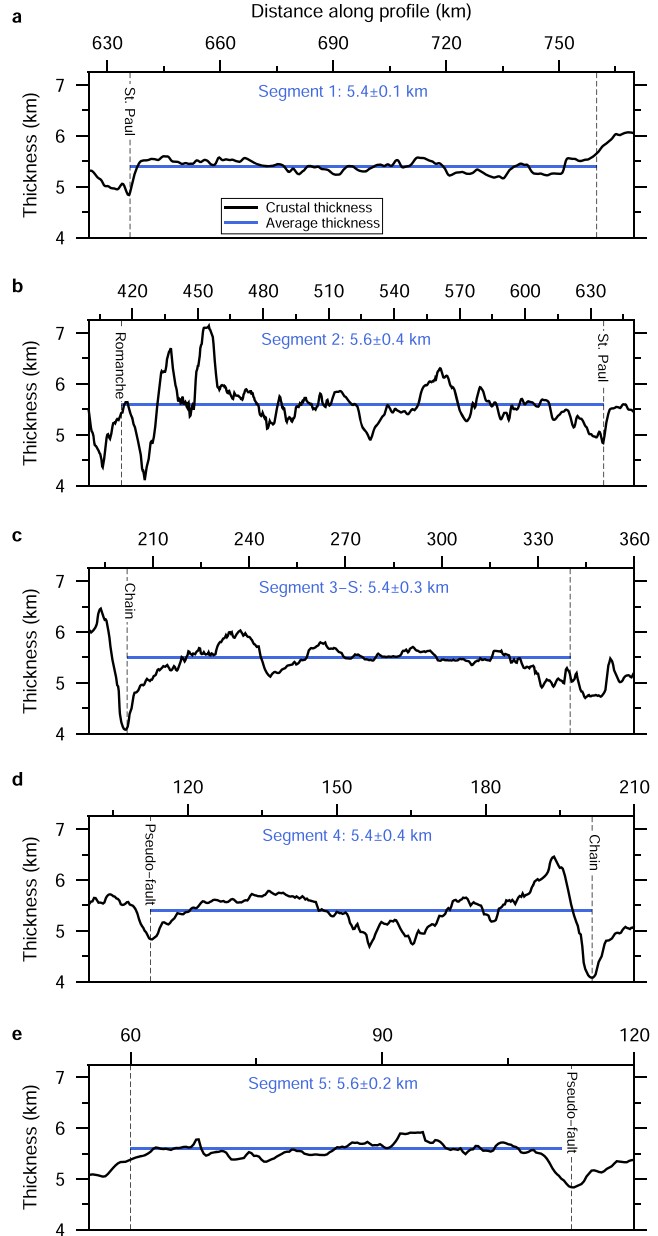

**Fig. 5 | Crustal thickness.** Crustal thickness variation (black curves) and average crustal thickness (blue lines) of each ridge segment. The dashed lines in each panel indicate the boundaries of ridge segments. The segment numbers, the average crustal thickness, and the standard deviation are marked in blue.

average, which is only ~500 m thinner than the average of Segment 2 to the north (Fig. 5b). Though the crustal Vp/Vs ratio is not constrained beneath the Romanche TF (Fig. 3c), Gregory et al.[45] argued that the thick crust at the Romanche transform zone is likely composed of fractured mafic rocks, and hence can support super-shear rupture during the 2016 $M_w$ 7.1 earthquake[51]. The crust beneath the ~10 km-wide Chain FZ trough has an average thickness of ~4.7 km, which is ~700–800 m thinner than the averages of the adjacent Segments 3-S and 4 (Fig. 5c, d). The thinnest crust within the Chain FZ trough is ~4.1 km, which is ~500 m thinner than that measured in ref. 46. The average crustal thickness within the pseudo-fault trough is ~5.2 km with the thinnest crust of ~4.8 km thick, which is ~200-400 m thinner than the adjacent segments (Fig. 5d, e). The crustal Vp and Vs (Fig. 3a, b) are lower at the FZ, TF and pseudo-fault regions relative to the surrounding crust, which could be due to the presence of fault-related damaged zones and fluids.

We take the standard deviation of mean crustal thickness as a measure of the intra-segment crustal thickness variation[13,14,25,52,53]. The standard deviations of the mean crustal thickness within these five segments are 0.1–0.4 km (Fig. 5), which is much smaller than those (≥1.0 km[13,14]) for other MAR segments where large along-axis crustal thinning is observed. However, they are very close to those (≤0.3 km[52,53]) observed along the fast-spreading EPR, suggesting relatively uniform crustal thickness within these segments. There is no evidence supporting thicker crust at the segment centre and greater than 2.8 km gradual crustal thinning towards the tectonic discontinuities within any of the five segments. The relatively low crustal Vp at the FZ, TF and pseudo-fault regions (Fig. 3a) indicates the thick crust at these tectonic discontinuities is not due to the inherent positive velocity-Moho depth trade-off in the travel time tomography[48]. These results are consistent with independent studies of these discontinuities[43,45,46], confirming that the crust beneath these discontinuities is not thin, and is of magmatic origin. A Monte–Carlo based uncertainty analysis yields the variance in the Moho depth of <400 m (see Methods; Supplementary Fig. 8b), demonstrating that the observed uniform crust does not result from a particular starting Vp model used in tomography. The checkerboard tests demonstrate that the used tomography method can recover the crustal thickness and its lateral variations for most portions along our seismic profile if the real variations in crustal thickness are ~2.5–3.0 km (see Methods; Supplementary Figs. 9–11). These tests demonstrate that the observed uniform crust within these five ridge segments is real.

## Discussion

Most previous studies using gravity and seismic data collected on the slow-spreading MAR reveal a systematic and substantial along-axis reduction in crustal thickness from the centre of segments towards the oceanic TFs, FZs and NTOs[3,8–18,21,22], suggesting a 3-D plume-like mantle upwelling[3,8] and/or a focused melt concentration to segment centres[13,19,20] beneath slow-spreading ridges. However, the five crustal segments with predominantly magmatically accreted crust in the equatorial Atlantic Ocean show slightly thin crust with little intra-segment crustal thickness variation throughout each segment (Figs. 1 and 5), which is remarkably different from the most previous observations in the Atlantic Ocean but similar to some recent observations on 6.6-61.2 Ma oceanic crust at 31°S in the South Atlantic Ocean[25]. The crust within these segments has either (1) been modified after crustal formation by tectonic extension and stretching during the amagmatic period, or (2) by the second stage of crustal accretion at RTIs, or (3) was originally formed uniformly at the ridge axis. Here, we discuss these 3 hypotheses and propose the best mechanism that explains the formation of the uniformly thick crust in the equatorial Atlantic Ocean.

Tectonic extension and stretching through normal faulting account for ~10% of plate separation at slow-spreading ridges[54,55]. The spacing and heave of normal faults are generally larger at segment ends than at segment centres, indicating that more tectonic extension occurs at segment ends due to decreasing magma supply[56,57]. Since the tectonic extension and stretching could thin the oceanic crust[54,55,58], more tectonic extension and stretching towards the segment ends will enhance the along-axis variation in crustal thickness[10,56], rather than making the crust uniform on the segment scale, validating that the tectonic extension and stretching cannot explain the uniform crust observed along our profile, ruling out the first hypothesis.

Some recent studies indicate that the crustal thickness at oceanic TF can be augmented by the second stage of magmatic accretion at RTIs[46,59] by dikes that are laterally emplaced into the transform valley at segment ends[46], forming J-shaped structures on the seafloor and thickening the crust. However, the gravity data reveal a ~5-km-thick crust in the western portion of the Chain transform zone[60], which is comparable with the average ~5.1 km-thick crust within the Chain FZ trough along our profile (Fig. 5c). Although we do not have any

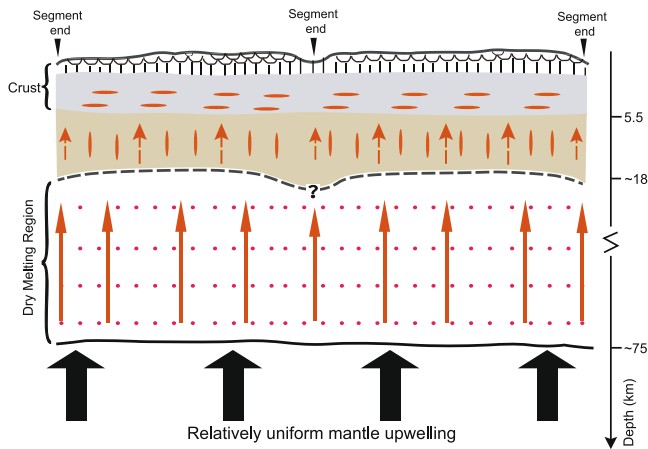

**Fig. 6 | Schematic diagram.** Schematic diagram showing relatively uniform crustal accretion and two-dimensional sheet-like mantle upwelling. The black thick arrows represent deep mantle upward flow. The black curve represents the start of dry melting. The red solid arrows represent melt migration in the dry melting region. The grey dashed curve represents the top of the melting zone. The red dashed arrows represent melt migration below the mantle-crust boundary. The depth of the top of the dry melting region is from ref. 80 and the presence of melt sills in the lower crust is determined in ref. 81. In this model, the melt is not focused to segment centres at the base of the lithosphere[19,20], but migrates upward vertically.

information about the crustal thickness for the Romanche FZ, the crust within the Romanche TF has similar thickness to those within the St. Paul and Chain FZs along our profile (Fig. 5). These observations suggest that the second stage of magmatic accretion may only slightly thicken the TF crust at RTIs in the equatorial Atlantic region. Furthermore, the J-shaped structures have very short extensions over the old ocean floor (generally <10 km)[59], suggesting that the second-stage of crustal augmentation occurs within a limited region in the vicinity of the RTI. Finally, for the lateral melt injection towards RTIs to occur during the second stage of crustal accretion, there should be sufficient melt from the mantle at segment ends. Collectively, the second-stage of magmatic accretion at RTIs could locally increase the crustal thickness at the RTI but is not likely to substantially change the crustal thickness within a >200 km long ridge segment between two oceanic TFs in the equatorial Atlantic Ocean, unless there is an abundant magma supply at segment ends.

Therefore, we propose that the observed wide-spread uniform crust along our profile was originally formed at the ridge axis (hypothesis 3) and that melt is nearly equally distributed all along the axis, as is observed for fast-spreading ridges. Below we analyse the patterns of mantle flow and melt migration along the ridge axis needed to form a uniform crust along the slow-spreading equatorial MAR system.

The absence of along-axis variations in the crustal thickness[61] at the slow-spreading (~10 mm/yr half-spreading rate) Reykjanes Ridge in the North Atlantic Ocean, where the crust is 8–10 km thick, has been interpreted to be due to a rapid ductile flow within the hot lower crust because of the influence of the Iceland hotspot[7]. Though the ridge between 5°N and the St. Paul TF is suggested to be influenced by the Sierra Leone plume[62,63], crustal thicknesses north and south of the St. Paul FZ are similar, ruling out any significant influence of a thermal anomaly on crustal accretion in our study area. The crust formed at the MAR in the equatorial Atlantic Ocean is much thinner (5.4–5.6 km) than that formed at the Reykjanes Ridge, indicating that the mantle is colder, and therefore the lower crust is not hot enough to enable rapid ductile flow within the lower crust. On the other hand, 3-D diapiric mantle upwelling would produce a large variation in crustal thickness within a ridge segment[3,8], which is inconsistent with our observations of the five crustal segments studied here. Therefore, the relatively

uniform crustal thickness observed in the equatorial Atlantic Ocean suggests a relatively uniform (nearly 2-D) mantle upwelling beneath the ridge axis at the time of the crustal accretion (Fig. 6).

The petrological studies of basaltic glasses dredged along the equatorial MAR between the St. Paul and Charcot TFs show a long-wavelength trend (~600 km) in the mantle potential temperature, the mean degree of partial melting and the maximum depth of the decompression melting. This long-wavelength trend seems to be independent of the offset and the location of TFs[62,63]. North of the St. Paul TF up to ~5 °N, the mantle potential temperature, the initial depth and the mean degree of mantle melting have little variation over ~400 km lateral distance in spite of the influence of the Sierra Leone plume[62,63]. These observations suggest that the extent of mantle melting and mantle isotherms are not significantly suppressed approaching TFs and NTOs as required by 3-D diapiric mantle upwelling[3,8], supporting relatively uniform mantle upwelling in our study region.

The lithosphere-asthenosphere boundary beneath slow-spreading ridges is expected to progressively deepen from the segment centre towards the oceanic TFs due to the juxtaposition of the ridge axis with cold lithosphere ('cold edge effect')[64], which could focus melt to the centre of the ridge segment[13,19,20] and produce a thick crust[9,12–18]. However, Wang et al.[47] found that the base of the lithosphere at the Romanche TF is uplifted by low-temperature hydrous mantle melting due to the presence of water. Meanwhile, the lithosphere beneath the ridge axis in the equatorial Atlantic Ocean is imaged to be thickened due to the enhanced hydrothermal cooling[65,66]. The combined effect of these processes could significantly decrease the cold edge effect of large oceanic TFs, leading to a relatively flat isotherm (and lithospheric base) along the ridge axis.

The basaltic glasses dredged from the vicinities of the western RTI of the St. Paul TF, the eastern RTIs of the Romanche and Chain TFs show large local geochemical and isotopic dispersions, suggesting the melt did not pool and mix in the magma chamber but segregated and erupted rapidly from the mantle[62]. This implies that the melt migrates vertically and rapidly beneath the ridge axis in the equatorial Atlantic Ocean region (Fig. 6), rather than focusing to the segment centres before eruption.

Along-axis variations in the seafloor depth and the Mantle Bouguer anomaly (MBA) indicate different modes of accretions along the MAR[67]. Here we compare the variations from the Lucky Strike segment at 37°N with the ridge segment between the St. Paul and Romanche TFs (Supplementary Fig. 13) to demonstrate the two different modes of crustal accretion and mantle upwelling. The Lucky Strike segment is characterised by a central volcano, indicating enhanced and focused magmatic accretion at the centre of the segment[68,69]. Consequently, large variations in seafloor depth (1.3–2.8 km) and MBA (~30 mGal) are observed from the centre to the distal ends of the Lucky Strike segment (Supplementary Fig. 13a, b). In contrast, the segment between the St. Paul and Romanche TFs shows <0.7–1.0 km variation in the seafloor depth and <15–20 mGal variation in the MBA (Supplementary Fig. 13c, d), much smaller than those observed along the Lucky Strike segment but comparable with those along fast-spreading ridges[3], supporting the presence of relatively uniform crust. Furthermore, the ridge segments in the equatorial Atlantic Ocean are characterised by wide and well-defined rift valleys (Supplementary Fig. 14), suggesting a low mantle temperature beneath these ridges ('colder segments')[67]. Using the seafloor bathymetry and MBA data, Thibaud et al.[67] have demonstrated that the 'colder segments' on the MAR generally show small variations in seafloor depth and MBA along ridge axis, which is consistent with our results and further supports our observation of uniform crustal accretion along the equatorial MAR.

The tectonic fabric of the ocean basin also reflects the different magmatic accretion patterns (focused or uniform) along the MAR. Though NTOs are observed on the ridge axis in the equatorial Atlantic

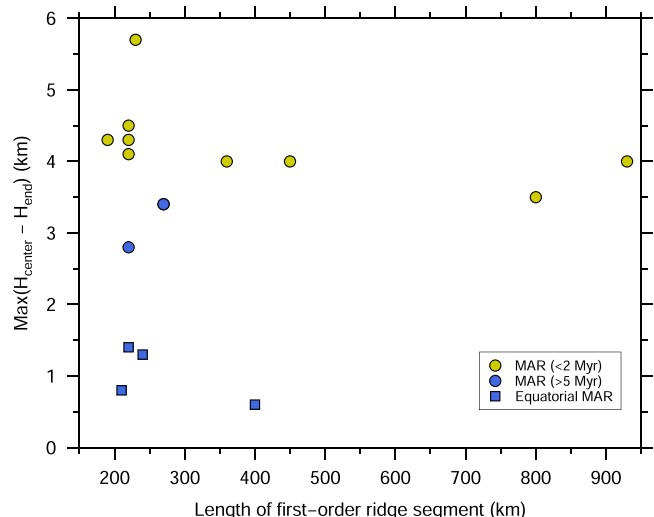

**Fig. 7 | Maximum crustal thickness variation versus ridge segment length.** Maximum crustal thickness variation between centres and ends of second-order ridge segments versus the length of the corresponding first-order ridge segment in the Atlantic Ocean. The crustal thickness constrained by both active-source seismic data and gravity data is included. The blue squares show results from this study for the equatorial Atlantic. Data sources are given in Supplementary Table 3.

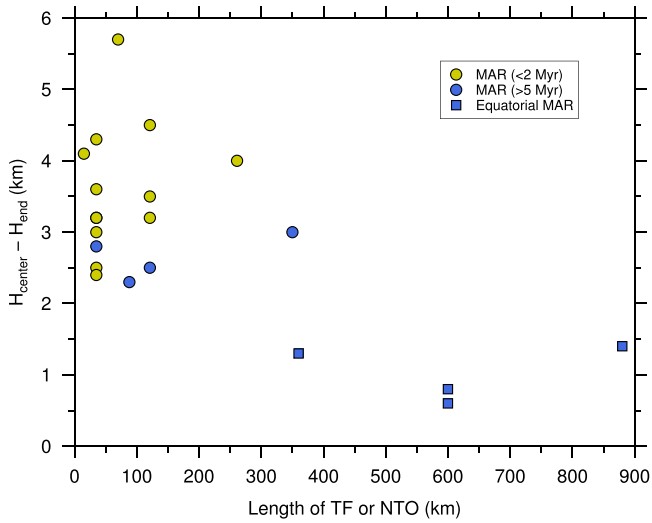

**Fig. 8 | Crustal thinning versus length of tectonic discontinuities.** Crustal thinning towards oceanic transform fault (TF) or non-transform offset (NTO) in the Atlantic Ocean as a function of the length of TF or NTO. Only crustal thickness constrained by active-source seismic data is considered. The blue squares show results from this study for the equatorial Atlantic. For a TF composed of several intra-transform faults, the total length of the TF is used. Data sources are given in Supplementary Table 4.

Ocean[44,46], the seafloor fabric[44] indicates that these NTOs are more transient features with nearly no imprint away from the ridge axis, suggesting that the base of the lithosphere beneath these short-lived NTOs does not deepen much and remains nearly flat, hindering melt focusing towards segment centres.

A nearly 2-D sheet-like mantle upwelling together with nearly vertical melt migration and rapid melt eruption would facilitate the formation of relatively uniform crust along the equatorial MAR at the time these five crustal segments were formed. Relatively uniform mantle upwelling has been inferred from the petrological observations along the present-day ridge axis in the equatorial Atlantic Ocean, and

our results indicate that this process has persisted over the last 70 Myr. Relatively 2-D uniform mantle upwelling in the equatorial Atlantic region differs from the previously proposed 3-D plume-like mantle upwelling model for slow-spreading ridges[8] based on observations from the North and South Atlantic Ocean, and instead is more similar to the 2-D sheet-like mantle flow occurring at the fast-spreading ridges[3].

We first investigated whether the pattern of mantle upwelling beneath slow-spreading ridges is related to first-order ridge segment length. We compiled the maximum crustal thickness variation between the centre and ends of second-order ridge segments and the length of the corresponding first-order ridge segments in the Atlantic Ocean (Fig. 7). Our compilation demonstrates that there is no positive or negative correlation between first-order segment length and crustal thickness variation. For example, the ~220 km-long first-order segment between the Hayes and Oceanographer TFs (Supplementary Fig. 15) shows a maximum of 2.8–4.3 km along-axis crustal thickness variations over the 0–5 Ma crust[13,14,17] while the ~800 km-long segment between the Kane and Atlantis TFs (Supplementary Fig. 15) shows a maximum crustal thickness variation of 3.5 km along the ridge axis[8]. The lengths of the first-order ridge segments in the equatorial Atlantic Ocean are similar to those of the segments between the Hayes and Oceanographer TFs, but relatively uniform crust is observed. We, therefore, argue that the length of the first-order ridge segment has no influence on the crustal accretion process, hence on the mantle upwelling pattern.

The 2-D sheet-like mantle upwelling beneath the equatorial MAR may be associated with the large oceanic TFs in the equatorial Atlantic Ocean. The Romanche TF is the largest oceanic TF on Earth[1,70], and the St. Paul and Chain TFs are much longer than most TFs in the North and South Atlantic Ocean[70] (Supplementary Fig. 15). By considering a more realistic brittle mantle weakening, Behn et al.[71] argued that the thermal structure of an oceanic TF is much warmer than predicted from the half-space cooling model, which can better fit the depth of seismicity. Their models also show enhanced mantle upwelling and much thinner lithosphere along the oceanic TF, especially at the centre of the transform, than estimated in previous studies using simplified rheologic laws. The presence of *en échelon* large TFs could significantly enhance mantle upwelling in the equatorial Atlantic Ocean. The compilation of crustal thickness variations (Fig. 8) from active-source seismic data indicates that the magnitude of the crustal thickness variation is more scattered and larger when the adjacent TF or NTO is short but is smaller for longer TFs. This supports that a megatransform could facilitate stable 2-D sheet-like mantle upwelling and relatively uniform crustal accretion along slow-spreading ridges.

The 2-D sheet-like mantle upwelling and formation of uniform crust could also be facilitated by relatively higher $CO_2$ and $H_2O$ concentrations in the primitive melt in the equatorial Atlantic Ocean region as compared to the North and South Atlantic Ocean[72] (Fig. 9). Large amounts of these volatiles in the mantle will decrease the mantle solidus and increase the depth extent of the melting regime[73], leading to enhanced production of melt beneath spreading centres. Volatile-rich melt in the mantle would also decrease the viscosity of the mantle[74], facilitating mantle flow. The presence of a large amount of volatiles ($CO_2$ and $H_2O$) in the melt would also decrease the density of the melt, and hence the mantle would be more buoyant, leading to more 2-D sheet-like upwelling. But the average crustal thickness of the five magmatic segments in the equatorial Atlantic Ocean is thinner than the global average[28], which could be attributed to the relatively low mantle temperature in the equatorial Atlantic region[63].

Detachment fault accommodation of plate separation occurs along ~50% of the ridge axis in the North Atlantic Ocean[22,26]. However, along our seismic profile, the magmatically accreted crustal segments comprise >90% of the total profile length, which is much higher than the previous estimates for slow-spreading ridges, suggesting that

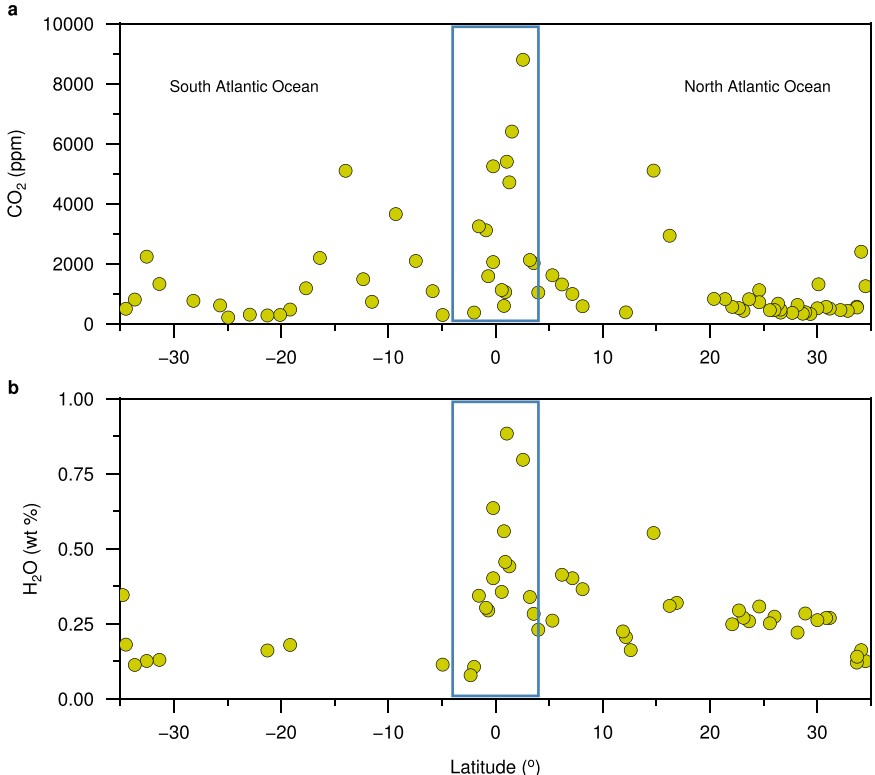

**Fig. 9 | CO₂ and H₂O concentrations along the Mid-Atlantic ridge (MAR).** Segment-averaged primary magma (**a**) $CO_2$ and (**b**) $H_2O$ concentrations along the MAR between 35ºS and 35ºN. The blue boxes outline our study region in the equatorial Atlantic Ocean. Data are from ref. 72.

mantle upwelling within a ridge-transform system plays an important role in the mode of crustal accretion along spreading centres. Our study indicates that crustal accretion and mantle upwelling along MORs within a ridge-transform system are significantly influenced by oceanic TFs offsetting ridges and by the volatile content in the upwelling mantle, and are more complex than being primarily spreading-rate dependent.

## Methods

### Seismic data analysis and travel time picking

The travel times of crustal refractions (Pg), wide-angle reflections from the Moho (PmP) and the mantle refractions (Pn) have been hand-picked on the seismic data after band-pass filtering (4–15 Hz for Pn arrivals and 4–20 Hz for Pg, PmP and S-wave arrivals). The Pg phases are identified as first arrivals outside of the water wave on 50 OBSs from ~4 to ~35 km offsets (Supplementary Fig. 1, orange dots). The picking uncertainty of the Pg arrivals varies between 30 and 50 ms. The Pn phases are identified from ~15–35 to ~250 km offsets for most OBSs[47]. In this work, only the Pn arrivals within 40 km offset are shown (Supplementary Fig. 1, blue dots). The picking uncertainty of the Pn arrivals varies between 45 and 100 ms. The PmP phases reflected off the Moho are identified at ~10–40 km offsets as second arrivals after the Pg and Pn phases (Supplementary Fig. 1, cyan dots). The picking uncertainty of the PmP arrivals is 50 ms or 70 ms.

The crustal and mantle S-wave arrivals are observed on seismic data recorded by 43 OBSs as secondary arrivals after the P-wave first arrivals and codas (Supplementary Fig. 2). In this study, we use Sg, SmS and Sn to represent the S-wave crustal refractions, Moho reflections and mantle refractions, respectively. We first pick the S-wave arrivals that have high signal-to-noise ratios using the trace-to-trace coherency of phases and then we invert these picks to obtain a smooth velocity model for the crust and uppermost mantle. Synthetic travel times of S-wave phases are modelled using this smooth velocity model, which is then used as a guide to identify and pick more S-wave arrivals that have

low signal-to-noise ratios[25,75]. The Sg arrivals are identified and picked between 5 and 35 km offsets (Supplementary Fig. 2, orange dots) with picking uncertainty ranging from 30 to 60 ms. The SmS phases are picked at offset ranges similar to the PmP arrivals (Supplementary Fig. 2, cyan dots), with picking uncertainty of 60–80 ms. The Sn arrivals are picked on 38 OBSs (Supplementary Fig. 2, blue dots) with picking uncertainty of 50–100 ms.

### Identifying the P-to-S and S-to-P conversion interfaces

The identified and picked S-wave seismic arrivals are characterised by an apparent velocity ≤4.0 km/s outside the direct water wave cone, indicating that these arrivals are S-wave arrivals. As the most portion of the seafloor along the seismic profile is covered by sediments, we use three different methods to identify the conversion interfaces of the picked S-wave arrivals:

(1) Comparisons of the four components (pressure from hydrophone and three geophone components) of the OBS data show the picked S-wave phases on the pressure component are also recorded on the vertical geophone component, but are not recorded on the horizontal geophone components (Supplementary Fig. 3). Though the orientations of the horizontal component geophones are not well-constrained, the absence of the identified S-wave arrivals on both horizontal components demonstrates that the picked S-wave arrivals have propagated as P-waves through the sediment (if present) to OBSs at the upward propagation leg (from crust to sediment or water);

(2) The earliest visible S-wave arrivals of strong energy on the horizontal component data is 1.0–1.6 s later than the picked S-wave arrivals on pressure data (Supplementary Fig. 3). This observation demonstrates that the difference between P- and S-wave velocities of the sedimentary layer can lead to 1.0–1.6 s travel time difference for signal with the same propagation mode in the crust and mantle. However, roughly linear regression analyses shows that the difference between intercepts of the Pg

and Sg travel times is <0.35 s (Supplementary Fig. 4), after subtracting the propagation time in water column. The intercept obtained from linear regression fit represents the propagation time in sediments, hence the small difference in travel time intercepts demonstrates the picked crustal and mantle Pg and Sg arrivals have the same propagation mode (P-wave propagation) in the sediments on both the downward propagation leg (from water to sediment or crust) and upward propagation leg (from crust to sediment or water). The small difference between Pg and Sg travel time intercepts can be attributed to the different ray paths of Pg and Sg arrivals in the sediment layer.

(3) The picked S-wave arrivals are the first visible S-wave of strongest amplitudes after the P-wave arrivals on the pressure data (Supplementary Fig. 2). Waveform modelling (Supplementary Fig. 5) using a simple layered model comprising an 1 km-thick sedimentary layer indicates that the crustal and mantle S-wave arrivals with P-to-S and S-to-P conversions at the sediment-basement interface exhibit the strongest amplitudes and shortest travel times on the pressure data among all the S-wave phases propagating through crust and mantle. This demonstrates that the picked S-wave arrivals have the P-to-S and S-to-P conversions at the sediment-basement interface for the region covered by sediments (Supplementary Fig. 5a).

Collectively, the picked S-wave arrivals have the P-to-S and S-to-P conversions at the sediment-basement interface for regions covered by sediments and at the water-basement interface for region free of sediment.

## Starting velocity models for travel time tomography

The seafloor depth and sediment thickness along the seismic profile are determined using coincident high-resolution bathymetry data and seismic reflection data[45,46]. The velocities of water and sediment are set to 1.5 km/s and 1.86 km/s in tomography[45,46], respectively. The initial crustal Vp model is constructed using a simplified one-dimensional (1-D) velocity profile[45,46] hanging from the basement. We introduce a smooth Moho interface into the starting Vp model at ~5.5 km depth below the basement. The initial structure of the Moho is obtained by smoothing the basement reflector within a 12 km-wide sliding window. The starting mantle Vp model is constructed utilising a 1-D velocity profile hanged from the smooth Moho, where the mantle Vp increases from 7.8 km/s just below the Moho with a vertical velocity gradient of $0.014 s^{-1}$ to 20 km sub-Moho depth. The model is discretized by 300 m grid spacing horizontally and 60 m grid spacing vertically. In this study, we first model and invert the picked travel times of P-wave arrivals to obtain the crustal and mantle Vp. The starting Vs model for inverting the picked travel times of S-wave arrivals is converted from the final Vp model, assuming a Vp/Vs ratio of 1.74 for both crust and mantle. The depth of the Moho constrained by the PmP travel times is fixed during the inversion of the travel times of S-wave arrivals.

## Ray-based travel time modelling and tomography

We model and invert all the picked travel times using a two-dimensional ray-based travel time tomography method[2,48]. The ray paths and travel times of the crustal and mantle arrivals are calculated using the shortest path method[76]. We take a top-down inversion strategy in the tomography of P-wave arrivals. The Pg arrivals are inverted first to constrain the velocity of the upper crust, followed by a joint inversion of Pg and PmP arrivals simultaneously to constrain the crustal velocity and the Moho depth. After the joint inversion of Pg and PmP arrivals, we fix the crustal velocity and the Moho depth, and invert the travel times of Pn arrivals only to update the mantle velocity. First- and second-order velocity regularisations are imposed to obtain a smooth velocity model[48]. The weight given to the horizontal derivatives is four times of that given to the vertical derivative. The

regularisation parameters are tested and selected in each iteration step to avoid the introduction of artefacts. We use the standard $\chi^2$ value[48] to measure the mismatch between the modelled and manually picked travel times. Large regularisation values are used at the early stage of tomography and the regularisation values are reduced when $\chi^2$ value approaches 1. We stop the inversion when the $\chi^2$ value approaches 1.0 or when non-physical artefacts appear when further decreasing the $\chi^2$ value. The final $\chi^2$ value is 1.6 for Pg and PmP arrivals (Supplementary Fig. 6a) and is 2.2 for Pn arrivals (Supplementary Fig. 6b). The final RMS misfits[48] are 42, 65 and 74 ms for Pg, PmP and Pn arrivals, respectively.

The same inversion method and strategies are used in the S-wave tomography, except the Moho depth constrained by the PmP arrivals is fixed in the joint inversion of Sg and SmS arrivals. Doing so, we have assumed that P- and S-waves share the same boundary between crust and upper mantle. The final $\chi^2$ value of combined Sg and SmS arrivals is 1.2 (Supplementary Fig. 6c) and that of Sn arrivals is 1.5 (Supplementary Fig. 6d). The final RMS misfits are 51, 63 and 98 ms for Sg, SmS and Sn arrivals, respectively.

## Ray coverage

We use the derivative weight sum (DWS)[77] to represent the density of ray coverage through the final tomographic models. The DWS of the P-wave arrivals throughout the crust and upper mantle is shown in Supplementary Fig. 7a. The Vp of the upper crust is constrained by dense rays of Pg arrivals between 60 and 760 km horizontal distances along the profile, while the lower crust has relatively sparse ray coverage constrained by PmP arrivals only. In the mantle, the Pn arrivals sample the upper mantle down to ~60 km depth below sea level beneath the Romanche transform valley and the maximum sampled depth gradually decreases to south and north[47], but here we only show the model down to 12.5 km depth (Supplementary Fig. 7a). Supplementary Fig. 7b shows the DWS of S-wave arrivals through the crust and mantle. There is good ray coverage throughout the crust between the horizontal distance 50 to 385 km and 460 to 760 km. The crustal Vs within the Romanche transform zone is not constrained.

## Monte-Carlo analysis

To assess the accuracy of the crustal and mantle velocities and the Moho depth, we perform Monte−Carlo analyses[78] starting from different initial models to produce different inverted models and estimate the standard deviation of these models from the mean model to measure the model variance. Here we only describe the details of the Monte−Carlo analysis for crustal Vp and Moho depth. We create 50 starting crustal models by randomly perturbing the minimum and maximum velocity of the one-dimensional crustal velocity profile by ±5% and the initial crustal thickness by ±750 m. These starting crustal models are then inverted using the same tomography algorithm as described before. We also vary the regularisation parameters on the velocity structure and Moho topography. The 50 final inverted models show similar crustal Vp structure, suggesting the inversion of the picked Pg and PmP travel times is robust. The variance in the final crustal Vp model is less than 0.1 km/s in the upper crust and is less than 0.3 km/s in the lower crust (Supplementary Fig. 8a). The maximum standard deviation of the Moho depth is ~400 m (Supplementary Fig. 8b). The preferred Moho (Fig. 3a) falls in the standard deviation of the average Moho depth from the Monte−Carlo analysis (see blue curve in Supplementary Fig. 8b). Similar Monte-Carlo analyses are performed to assess the variance in the crustal vs. Supplementary Fig. 8c shows the variance of the crustal Vs calculated using 50 final inverted models. For most portion of the crust, the crustal Vs has a variance <0.1 km/s, and large variances >0.1 km/s are observed around the TF and FZs and at the southern and northern extremity of the model.

## Checkerboard tests

We also use the checkerboard test described in ref. 4 to assess the resolution of the final crustal Vp and Vs models. We first perform the checkerboard test to assess the crustal Vp. The checkerboard input models are designed by adding 2-D sinusoidal velocity perturbation of ±10% into the starting velocity model after slight smoothing. The anomaly size in the checkerboard models is 20 km × 2 km (Supplementary Figs. 9a, d, 10a, d and 11a, d). The checkerboard patterns with opposite polarities are used to test whether the resolution is independent on the polarity of the anomaly. A sinusoidal perturbation with a half-wavelength of 50 km (Supplementary Figs. 9a, d), 100 km (Supplementary Fig. 10a, d) and 200 km (Supplementary Fig. 11a, d) was added to the initial Moho to examine the resolvability in the Moho depth[4,46]. The amplitude of the sinusoidal perturbation in the Moho depth is 1.25 km, making the crustal thickness variations in the checkerboard models be ~2.5–3.0 km (red curves in Supplementary Figs. 9c, f, 10c, and 11c, f). Synthetic seismic travel times are calculated using the checkerboard models and the same receiver-source geometry as the Pg and PmP picks. Some random noise is added to these synthetic picks to represent the picking uncertainty. We invert these synthetic travel times using the same workflow as that for the OBS dataset, starting from the same initial model. The results show that velocity anomalies of size 20 km × 2 km are almost completely recovered (Supplementary Figs. 9b, e, 10b, e and 11b, e), and the recovery is independent on the polarity of the anomaly. The crustal thickness and its lateral variation for most portions along our seismic profile are recovered after tomography (black curves in Supplementary Figs. 9c, f, 10c, f, and 11c, f). A similar checkerboard test is applied to assess the resolution of crustal Vs model. The checkerboard test shows a higher resolution of the crustal Vs model than the crustal Vp model, where the 8% velocity anomaly with the 15 km × 3 km size can be well recovered (Supplementary Fig. 12).

## Data availability

The high-resolution bathymetry data, the multichannel seismic reflection data and the OBS data from the OBS26 to OBS50 are available online (https://doi.pangaea.de/10.1594/PANGAEA.922331) under the condition of acknowledging Marjanović et al., 2020 (https://doi.org/10.1029/2020JB020275). The OBS data from the OBS01 to OBS15 are available online (https://doi.pangaea.de/10.1594/PANGAEA.937195) under the condition of acknowledging Growe et al., 2021 (https://doi.org/10.1029/2021JB022456). The OBS data from the OBS16 to OBS25 are available online (https://doi.org/10.1594/PANGAEA.946565) under the condition of acknowledging Wang et al., 2022 (https://doi.org/10.1038/s41561-022-01003-3).

## Code availability

The travel time tomography code used in this work is propriety code but will be available on request from the second author S. C. Singh (singh@ipgp.fr).

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

## Acknowledgements

We are grateful to the officers, crew, and scientific technicians of the 2018 ILAB-SPARC cruise for their hard work. The research leading to these results has received funding from the European Research Council under the European Union's Seventh Framework Programme (FP7/2007–2013)/ ERC Advance Grant no. 339442 TransAtlanticILAB. Thanks go to Marcia Maia for valuable discussions on the interpretation of gravity data. Zhikai appreciates Milena Marjanović, Emma P.M. Gregory, Mathilde Cannat, Venkata Vaddineni, Kevin Growe and Jie Chen for their help and useful discussions. Some results presented in this paper are performed on the S-CAPAD platform of Institut de Physique du Globe de Paris (IPGP), France. This is an IPGP contribution number 4273.

## Author contributions

Z.W. processed and analysed the data and wrote the paper. S.C.S. developed the project, led the data acquisition and supervised the data processing, interpretation and writing.

## Competing interests

The authors declare no competing interests.
