## [Peer Review File · Nature Communications]

REVIEWER COMMENTS

Reviewer #1 (Remarks to the Author):

Dear authors,

the manuscript is nicely written and presents a part of recently acquired wide-angle seismic data along an ~870 km long transect imaging oceanic crust of different ages between 5° N and 5° S. An S-wave analysis adds to the database of the area. P-wave traveltimes are the same as in the two previous works (44, 45) and the accompanying paper. But the P-wave model was generated newly and used for newly generated S-wave tomography. The authors propose a uniform crustal accretion caused by 2D sheet-like mantle upwelling similar to fast spreading ridges to occur at the slow-spreading equatorial Mid-Atlantic Ridge (MAR) which is different to observations further north and south along the MAR. The figures are presented well and have a high quality. The data analysis and methods are presented in a good standard and good details are given to reproduce the work.

Based on observations in reasonably old oceanic crust (8-70 Ma), the authors conclude on the style of accretion during the formation of the crust. Which is also targeted by the title. I'm missing an explanation why we see differences to the northern and southern MAR. The authors state that their findings contradict previous works at the northern and southern MAR. On the other hand, most named works observe young oceanic crust <2 Ma, while the authors of this work present oceanic crust >7 Ma. Why do we have uniform crustal accretion along the segments at the equatorial MAR but not in the before mentioned regions north or south of 5°. Indeed, the data image old oceanic crust and a direct comparison to today's crustal accretion specifically at these 5 segments would be necessary to conclude on the source of the observed uniform crust.

The authors name three hypotheses that should explain the reason for this uniform crustal structure. Their discussion is announced, I however can hardly find it completely discussed. Hypothesis 2 has been proposed in previous works from co-authors (45, 56). While hypothesis 1 is widely discussed and even further explanations are given that accretion was uniform during crustal accretion. However, the other two hypotheses are almost not discussed and it is not explained why they should not be valid. I do not see why hypothesis 2 is ruled out (two-stage accretion). What arguments contradict a two stage crustal accretion, during the time of segment formation and during the time the aging crust is passing the adjacent ridge axis?

In my opinion additional evidence is needed to support the interpretation and conclusions and to rule out the other named hypotheses. While I still find the work interesting and worth publishing after minor changes, I think there are other journals this work would fit better, since it does not present a very novel and clear explanation or extremely novel findings. The dataset is presented in many pieces.

Detailed points:

Lines 55-60: Give references to the both modes.

Lines 73/74: Give a reference.

Line 85: Define the area of "equatorial Atlantic Ocean". Is this according to fig. 2 it is +/- 5°?

Line 118: Make clear that these are converted (PS) or since hydrophones were used for picking PSP converted. What is the conversion horizon? For young crust with no sediments most often it is the seafloor itself.

Line 134: Does tectonic accretion need a detachment fault? I understood that detachments develop under certain conditions regarding the magmatic budget. Too less or too much magma, no detachment development. Compare to Buck et al., 2005. M=Magmatic Budget; $M < 0.5$

Line 143: add PmP  "... reflections (PmP)."

Lines 161-163: It looks like you observe a seismic Moho in that segment. What does this

reflector stand for? A really seismic Moho separating igneous crust from mantle or a serpentinisation front or any other explanation?

Line 178: "a slight thinning" <-- quantify slight. Or is this represented by 5.4 km total crustal thickness?

Line 181: Crust thins by ~600 m to a total crustal thickness of xx km? Does your total crustal thickness include sediments or not? In line 174 you state a total change in crustal thickness of 200 m. This contradicts line 181 with ~600 m change in crustal thickness.

Lines 227-231: The authors propose three hypotheses to explain the small variation in crustal thickness in old crust (>7 Ma). However, in the following sections the discussion of these three hypothesis concentrates on evidences for hypothesis 1 mainly. Hypothesis 2: Stretching here refers to the segment centre after crustal formation? I see this hardly discussed in the following sections. A stretching of the segment centre only can be ruled out as long as magnetic anomalies are ridge parallel. Hypothesis 3: I don't see this discussed in the following sections. Why do you rule out this 2 stage accretion?

Line 273: Remember that you are not studying the segment during the time of crustal formation (first stage).

Line 276: Did you look at the mantle bouger anomaly at the recent segment centres of this five segments studied in this work?

Lines 295-297: Give a reference for the "thick crust".

Lines 353-364: This seems to be a kind of doubling to the lines 328-331.

Line 384: 50% versus 90% magmatic accretion < You look at very different times of crustal accretion within the 5 segments. You might image magmatically active times of the specific segments.

Line 394: Why did you change frequency, was the signal for Pn so much improved but Pg and PmP haven't been visible any longer?

Lines 457 and followings: You observe two times converted S-Waves. Starting as P-Wave at the source, converting at or in the subsurface, travelling as S through the subsurface becoming converted to P at a boundary or seafloor and than being recorded as P-Wave to the hydrophone. Where does the conversion takes place? If at the seafloor, did you take the sedimentary cover into account? It will extremely delay the S-wave arrivals even if only a few centimeter of sediments.

Line 467: add km to "... 60 to 760 horizontal distances ..."

Line 468: change to something more clear maybe: "the lower crust has sparse ray coverage, PmP only."

Lines 476-477: You do not have Vs there at all.

Line 486: "50 models"  "50 calculated models" or "50 inverted models"

Line 512: "...models is 20x3 (Supp..." add "km". Did you also run a checkerboard test with inverted anomalies, to show that the resolution is independent of the polarity of the anomaly?

Lines 527-528: "higher resolution of Vs than Vp" <-- Would be expected because of low Vs wave speed and wave length.

Lines 529-531: This is a very weak explanation, because you do not allow for Moho depth adjustments. This does not mean that this Vp-Moho is the best possible. By this you do not allow for more variation.

Fig. 4: Panel b, segment 3N, shows models with a negative vertical velocity gradient which hints to problems in the inversion or how do you explain a negative velocity gradient in your models?

Suppl. Fig. 5: panel d, I do not see the advantage of presenting the variance of Vp/Vs, since Vp is kept constant it more or less reflects the variance of Vs.

Reviewer #2 (Remarks to the Author):

This manuscript presents a new Vp and Vs model for the oceanic crust across several segments in the equatorial Atlantic. The seismic line spans a large range of crustal ages. The velocity modeling shows that (1) the crust was magmatically accreted, and (2) crustal thickness is roughly constant, in contrast to other slow-spreading crust where crustal thickening is observed toward segment centers. The authors interpret this crustal thickness

observation as indicating that the crust in this part of the Atlantic accreted magmatically via 2D sheet-like mantle upwelling rather than the 3D plume-like upwelling generally hypothesized for slow-spreading ridges.

This is an interesting result, and is well presented in the manuscript. It has important implications for how we conceptualize crustal accretion and mantle upwelling at ridges. I do have some questions and comments, which are listed below. The main points that I think need addressing are these: (1) the resolution tests could do a better job of convincing readers that the uniform crustal thickness is a robust result; and (2) the manuscript makes some statements about the implications of the study that seem to equate magmatic upwelling at ridges with 2D sheet-like upwelling, although elsewhere the existence of 3D plume-like upwelling at some slow-spreading segments is discussed. I suspect the first issue is only because of the choice of testing parameters, and the second can be addressed by refining some of those over-broad statements.

I understand the desire not to speculate too much without solid evidence, but I would be interested to read more of the authors' thoughts on why magmatic crust in the equatorial Atlantic is apparently accreting differently than elsewhere along the MAR. Is it the presence of extra-long transforms and their stabilizing effects on ridge thermal structure that make this 2D upwelling possible in a slow-spreading regime? This is suggested in the discussion, but the abstract and final conclusions focus on the potential to map out 2D upwelling regions rather than ideas as to why this upwelling scenario exists. Discussion of the role of these long transforms seems especially relevant in the context of the related manuscript on the velocity structure beneath Romanche.

Specific comments:

L18: I am not sure that "contradicting [a] previous hypothesis" is accurate here. I assume that the "previous hypothesis" is supposed to be that 2D sheet-like upwelling cannot occur at slow-spreading ridges, though this isn't explicitly stated. Either that previous hypothesis should be more clearly defined (if it is actually a previous hypothesis and not a straw man), or I would suggest rephrasing this to state that "...uniform magmatic accretion at slow spreading rates is due to a two-dimensional sheet-like mantle upwelling more similar to magmatism at fast-spreading ridges than at previously surveyed slow-spreading ridge segments."

L212-214: This is slightly concerning – if this inversion could be missing as much as 1 km of crustal thickness variation, then it is difficult to argue that the crustal thickness here is unambiguously uniform (and, in particular, more uniform from ends to centers than other MAR crust – if your points on Fig 1 moved up 1 km on the vertical axis, they would start to cluster with the other MAR points). The final sentence of this paragraph is "These tests demonstrate that the observed uniform crust within these five ridge segments is real" (L215). Please clarify the justification for that last sentence.

Fig 6 and 7, L514-515: Related to the previous point, why was the crustal thickness perturbation wavelength for the checkerboard tests set at 70 km? Given that the recovered crustal thickness from that perturbation is significantly more uniform than the input model, it's not a very convincing argument for this model being able to properly resolve crustal thickness, but perhaps a longer perturbation wavelength could be better resolved. A longer wavelength would also be relevant to the question of whether the crustal thickness is uniform on a segment scale.

L275 paragraph: It seems like there are two arguments here mixed together: (1) uniform crustal thickness implies uniform (non-3D) upwelling, and (2) there are long-wavelength trends in MORB geochemistry that aren't segment-scale and don't correlate to transform faults, indicating that the presence of transforms isn't controlling mantle temperatures enough to make upwelling more 3D and diapiric. I think the geochemistry part should be a paragraph on its own. The first part is more of a restatement of the main argument of the paper.

L338 paragraph: This paragraph also seems to contain more than one big idea: (1) some kind of balance between TF enhanced melting and hydrothermal cooling at segment centers might lead to relatively flat isotherms and uniform upwelling (up to L353), and (2) the length of a transform (but not specifically the offset on the transform) might have some effects on thermal structure, with mega-transforms facilitating this 2D upwelling scenario. I would recommend separating the first part of the paragraph from the second part, and reordering the paragraph for the second part to put the main point upfront.

L386: I am not sure what "suggesting that the mode of accretion plays a more important role in defining the 2-D versus 3-D mantle upwelling" means here. Does "mode of accretion" refer to tectonic vs magmatic accretion? If this is intended to imply a connection between the (large) percentage of magmatically accreted crust in the equatorial Atlantic and the mode of upwelling, i.e. 2D upwelling at slow spreading rates is possible if most of the crust is magmatically accreted, then I'd like to see more evidence for that connection/discussion of some mechanism. Otherwise it sounds kind of like correlation implying causation.

L387-389 (and also L19-20): I am not convinced by this idea that the "lateral extent of magmatic accretion could be used to map the 2D sheet-like mantle upwelling regions along the global spreading ridge system." Maybe what is meant is that the lateral extent of uniform-thickness magmatic-style crust can be used to define where sheet-like upwelling has occurred? "Magmatic accretion" also encompasses crust with segment-scale thickness variations that was accreted via 3D plume-like magmatic processes at slow-spreading ridges.

Some picky comments on grammar/typos/figures:

L18: "contradicting previous hypotheses" or "contradicting the previous hypothesis" - as written, the singular "hypothesis" doesn't work.

L19: the lateral extent

L284-287: For the sentence that starts "While north of the St. Paul fracture zone..." I think if you remove the word "While" it would make sense grammatically.

L403: 50 or 70 ms? Do you mean 50-70 ms?

Supp. Fig 2: Why shift times by 2 seconds rather than having the time axis start at 2? It's not incorrect or anything, but I'm genuinely curious if there's a reason for doing this.

L467-468: for the 60 to 760 "horizontal distances" is the unit km along the seismic line?

Supp. Fig 6c and 7c: Could you use more distinct colors than red and magenta for the two Moho lines? I can't tell them apart.

Fig 1: How is the boundary between uniform and segment center-focused crust defined (dashed black horizontal line)? Is it just set at the smallest known thickness variation for non-equatorial Atlantic segments? Please add a citation or provide justification (or remove the line - not sure what purpose it serves since the groups of data points are already fairly distinct).

Fig 2: It would be useful to have a figure panel showing the (half-)spreading rate at the time of formation along this seismic line, so we don't have to try and match fig 2b to the mapped isochrons in fig 2a. This could replace 2b, or be an additional panel.

Fig 3, 5, etc: Why is there a gap between segments 2 and 3N? It seems odd particularly because the gap is narrower than the unconstrained part of the Vs model in the Romanche FZ, at least in the bars across the top of Fig 3a.

Figs 1 and 7: caption for 1 says only active source-derived crustal thicknesses are used, but the caption for 7 says data come from both seismic and gravity data; the number of points is the same for the two figures. Are any of them from gravity data?

Reviewer #3 (Remarks to the Author):

What are the noteworthy results?

Noteworthy is that the authors find relatively uniform crustal thickness at 5 off-axis ridge segments in the central Atlantic. In addition, 4.5 out of 5 segments appear constructed magmatically indicating two-dimensional mantle upwelling and crustal accretion.

Will the work be of significance to the field and related fields? How does it compare to the established literature? If the work is not original, please provide relevant references.

This result is surprising because previous results in both the northern and southern Atlantic ocean show:

- i. Frequent amagmatic crustal accretion at the MAR especially near major first and second order offsets.**
- ii. Significant observed variations in crustal thickness within MAR segments.**

Existing crustal thickness variations have motivated two models in the established literature: (i) three-dimensional plume-like mantle upwelling; and (ii) two-dimensional mantle upwelling with melt focusing to segment centers along the topography at the base of the lithosphere.

- The Discussion argues against both of these two models for the equatorial Atlantic.**
- I am under the impression that the Discussion also argues against these models more broadly in the paragraph comparing crustal thickness variations to 1st and 2nd order segment lengths (lines 315-336) for a compilation of slow spreading ridges. However, it is not quite clear what process this comparison is seeking to test. Clarify the purpose of the paragraph making this comparison.**

Does the work support the conclusions and claims, or is additional evidence needed?

This work is lacking in that a new model that explains both these new results and the existing observations is not clearly given in the abstract nor the conclusions.

Of notable relevance to developing a new model are the facts that:

- 1. this region of the mid-ocean ridge is broken up by very long offset transform faults and is a noted mantle cold spot (though this is confusing with additional discussion of a Cobb hotspot). This is not mentioned in the abstract.**
- 2. the equatorial Atlantic is a cold spot and this appears crucial to the final interpretation in the very last (concluding?) paragraph (line 374-375, line 378) .But adding this at the very end of the manuscript seems like a suddenly introduced new hypothesis. Specifically, this fact was not mentioned in the abstract and introduction.**
- 3. existing observations seems to support more drastic crustal thickness variations along active ridge segments than off axis. This is mentioned and models related to this observation are rejected, however, an explanation for this observation is not provided.**

In the Discussion, paragraph starting at line 275, a convincing argument is made that in the equatorial Atlantic the new crustal thickness measurements and existing geochemical observations support a 2D style of mantle upwelling and crustal accretion.

**The model that the authors appear to favor is buried in the Discussion (lines 362-364):
"this suggests the mega-transform could facilitate a stable 2-D sheet-like mantel upwelling**

and a relatively uniform crustal accretion". This seems to be the authors main conclusion and needs to be included in the Abstract. An outstanding question that I have about this process is whether the relatively cold mantle temperature in the region is adding to the effect of mega-transforms in controlling the mantle thermal structure.

Instead what the structure of the paper conveys to be more important is the poorly developed idea that: "the lateral exten[d]T of magmatic accretion could be used to define the sheet-like mantle upwelling regions beneath the global ridge system". It is simply stated at the end of the Abstract (note that when reading it I could not understand what "lateral extent" referred to). The next mention is in the very last sentence of the manuscript: "and the lateral extent of the magmatic accretion could be used to map the 2-D sheet-like mantle upwelling regions along the global spreading ridge system". Why is this important? This point does not seem as relevant to me as featuring the main interpretation/model in the Abstract.

The final paragraph (lines 386-387) also states: "that the mode of accretion plays an important role in defining the 2-D versus 3-D mantle upwelling". What exactly is the thinking here? Does the mode of accretion control whether mantle upwelling is 2-D versus 3-D, i.e. is the control top-down? Or is the mode accretion a consequence of the pattern of mantle upwelling, i.e. the control is bottom-up? I believe the second is being argued here – using language that is more clear than "defnining" might help clarify the intent of this statement.

Are there any flaws in the data analysis, interpretation and conclusions? Do these prohibit publication or require revision?

- The seismic data shown in the supplement looks good. There are very nice S wave arrivals – a consequence of collecting this data off-axis where acoustic waves can convert into S waves in the seafloor sediments at the ray entry point.
- Lacking is a specific test of whether a model with along axis variations in crustal thickness is precluded by the data. This is required to support the interpretation that crustal thickness is more or less uniform along the 5 ridge segments. Specifically, the variable ability to recover the Moho checkerboard pattern raises this question (lines 212-214 & Supplemental Figures 6-8). A discussion the relative weight of travel times for direct P phases versus those for reflected phases on the results is not included.
- The effect of varying the regularization parameters on the velocity structure and Moho topography is not explored in the Monte Carlo analysis.
- To support the suggestion that colder mantle is playing a role in generating an average crustal thickness of 5.5 km (lines 379-381) the paper needs to include a calculation that shows whether or not a 150°C reduction in mantle is consistent with a reduction in crustal thickness/melt production of 500 m.

Is the methodology sound? Does the work meet the expected standards in your field?

- The seismic analysis seems sound and uses a well-established method

Is there enough detail provided in the methods for the work to be reproduced?

- Yes. However, the choice of regularization parameters for the preferred model are not given.

Detailed comments:

- Abstract: The two existing models for crustal thickness variations at slow spreading ridges are incorrectly merged together in the phrase (lines 11-12) "due a three-dimensional plume-like mantle upwelling with melt focusing to segment centres".

- **Introduction – issues with the description of crustal accretion at fast spreading ridges:**
 - **Line 32:**
 - **Delete NTOs: Second order offsets are OSCs at fast spreading ridges and NTOs at slow spreading ridges**
 - **This statement ignores the literature that crust is not thin beneath OSCs at the EPR (e.g. Canales et al., 2003).**
 - **Lines 33-34: Incorrect: fast spreading ridges are fed by individual mantle upwellings (e.g. Toomey 2007) and are not just two-dimensional sheet-like mantle upwellings.**
- **Supplemental Figures 6 &7: Caption reads that this is for Vp model, figures are labelled as being S models**

In summary, I recommend major revisions so that the manuscript more clearly argues for the main processes that the authors infer from these new observations.

Reviewers' comments are shown in black; authors' response is shown in blue.

Reviewer #1:

The manuscript is nicely written and presents a part of recently acquired wide-angle seismic data along an ~870 km long transect imaging oceanic crust of different ages between 5°N and 5°S. An S-wave analysis adds to the database of the area. P-wave traveltimes are the same as in the two previous works (44, 45) and the accompanying paper. But the P-wave model was generated newly and used for newly generated S-wave tomography. The authors propose a uniform crustal accretion caused by 2D sheet-like mantle upwelling similar to fast spreading ridges to occur at the slow-spreading equatorial Mid-Atlantic Ridge (MAR) which is different to observations further north and south along the MAR. The figures are presented well and have a high quality. The data analysis and methods are presented in a good standard and good details are given to reproduce the work.

Based on observations in reasonably old oceanic crust (8-70 Ma), the authors conclude on the style of accretion during the formation of the crust, which is also targeted by the title. I'm missing an explanation why we see differences to the northern and southern MAR. The authors state that their findings contradict previous works at the northern and southern MAR. On the other hand, most named works observe young oceanic crust <2 Ma, while the authors of this work present oceanic crust >7 Ma. Why do we have uniform crustal accretion along the segments at the equatorial MAR but not in the before mentioned regions north or south of 5°. Indeed, the data image old oceanic crust and a direct comparison to today's crustal accretion specifically at these 5 segments would be necessary to conclude on the source of the observed uniform crust.

In the revised manuscript, we have proposed two mechanisms that are likely facilitate a 2-D mantle upwelling beneath the slow-spreading ridges in the equatorial Atlantic Ocean: (1) *en échelon* large oceanic transform faults (TFs) and (2) higher CO₂ and H₂O concentrations in the primitive mantle melt, observed along the equatorial MAR.

The spreading ridge in the equatorial Atlantic Ocean is offset by several large oceanic TFs. The Romanche TF is the largest oceanic TF on the Earth, and the St. Paul and Chain TFs are much longer than most TFs in the North and South Atlantic Ocean [Wolfson-Schwehr and Boettcher, 2019] (Supplementary Fig. 13). By considering a more realistic brittle mantle weakening, Behn *et al.* [2007] argued that the thermal structure of the transform zone is much warmer than that predicted from the half-space cooling model, which can better fit the depth of seismicity on the oceanic TFs. Their modelling also demonstrates an enhanced mantle upwelling along the transform zone and a much thinner lithosphere at the transform zone, especially at the centre of the transform, than estimated in previous studies using simplified rheologic laws, as shown by Wang *et al.* (2022, Supplementary Fig.

10). The *en échelon* large TFs could significantly enhance the mantle upwelling in the equatorial Atlantic Ocean.

The 2-D sheet-like mantle upwelling and the formation of uniform crust could also be facilitated by the relatively higher CO₂ and H₂O concentrations in the melt in the equatorial Atlantic Ocean region [Le Voyer *et al.*, 2019] (Fig. 9 in main text) as compared to the North and South Atlantic Ocean where large crustal thickness variations are observed. Large amount of these volatiles in the mantle will decrease the mantle solidus and increase the depth extent of the melting regime [Keller and Katz, 2016], leading to the enhanced production of melt beneath the spreading centres. Melt in the mantle can decrease the viscosity of the mantle [Whitehead *et al.*, 1984], which could facilitate the mantle to flow. The presence of a large amount of volatiles (CO₂ and H₂O) would also decrease the density of the melt and mantle, and hence the mantle would be more buoyant, leading to a more 2D sheet-like upwelling.

It would be great to have the crustal thickness estimation along the ridge axis, parallel to our profile, but that would be mega-project of its own, and is beyond the scope of this paper. Furthermore, most of the existing seismic studies of the axis of slow-spreading ridges have been limited to the length of the segments, not beyond the transform, and hence has poor constrains on the transform system.

The authors name three hypotheses that should explain the reason for this uniform crustal structure. Their discussion is announced, I however can hardly find it completely discussed. Hypothesis 2 has been proposed in previous works from co-authors (45, 56). While hypothesis 1 is widely discussed and even further explanations are given that accretion was uniform during crustal accretion. However, the other two hypotheses are almost not discussed and it is not explained why they should not be valid. I do not see why hypothesis 2 is ruled out (two-stage accretion). What arguments contradict a two stage crustal accretion, during the time of segment formation and during the time the aging crust is passing the adjacent ridge axis?

We proposed three hypotheses for the uniform crust of magmatic origin observed within the five segments in the equatorial Atlantic Ocean: the crust has either (1) been modified after crustal formation by tectonic extension and stretching during the amagmatic period, or (2) by a second-stage crustal accretion at RTIs, or (3) was originally formed uniformly at the ridge axis. We rule out the hypotheses 1 and 2 by the arguments in the second and third paragraphs in the 'Discussions' section.

The tectonic extension and stretching through normal faulting account for ~10% of plate separation at the slow-spreading ridges [Combiér *et al.*, 2015; Escartón *et al.*, 1999]. The spacing and heave of normal faults are generally larger at segment ends than those at segment centres of the slow-spreading ridges, indicating that more tectonic extension occurs at segment ends due to the decreasing magma

supply [Shaw, 1992; Shaw and Lin, 1993]. Since the tectonic extension and stretching could thin the oceanic crust [Combi *et al.*, 2015; Escartín *et al.*, 1999; Escartín and Lin, 1995], more tectonic extension and stretching towards the segment ends will enhance the along-axis variation in crustal thickness [Detrick *et al.*, 1995; Shaw, 1992], rather than making the crust uniform at segment scale, validating that the tectonic extension and stretching cannot explain the uniform crust observed along our profile, ruling out the first hypothesis.

Marjanović *et al.* [2020] and Grevemeyer *et al.* [2021] proposed that the oceanic transform fault (TF) crust can get augmented by a second-stage of magmatic accretion at the RTI, which accounts for the relatively thicker crust and relatively shallower seafloor at the oceanic fracture zone (FZ) than at the TF valley. However, the gravity data reveal a ~5 km-thick crust within the Chain transform zone [Harmon *et al.*, 2018] comparable with the average ~5.1 km-thick crust within the Chain FZ along our profile, suggesting that the second-stage of magmatic accretion may have limited augmentation to the TF crust at RTIs. Grevemeyer *et al.* [2021] argued that this second-stage of magmatic accretion at RTI forms the J-shaped ridges observed on the bathymetry data, which can extend across the RTIs and often terminate in the older plate. However, the J-shaped ridges have very short extensions (generally <10 km) over the old ocean floor [Grevemeyer *et al.*, 2021], suggesting that the second-stage of crustal augmentation occurs within a limited region in the vicinity of the RTI. Since the spreading centre between two major TFs is ≥ 200 km long in our study region, the second-stage of magmatic accretion at RTI cannot change the segment-scale variations in the crustal thickness. Therefore we rule out the hypothesis 2.

We have augmented these discussions in the revised version of the paper.

In my opinion additional evidence is needed to support the interpretation and conclusions and to rule out the other named hypotheses. While I still find the work interesting and worth publishing after minor changes, I think there are other journals this work would fit better, since it does not present a very novel and clear explanation or extremely novel findings. The dataset is presented in many pieces.

Detailed points:

Lines 55-60: Give references to the both modes.

We cited Escartín *et al.* [2008] and Searle [2013] for the magmatic accretion mode and cited Cann *et al.* [1997] for the tectonic accretion mode.

Lines 73/74: Give a reference.

We cited Carlson and Miller [1997] as reference.

Line 85: Define the area of "equatorial Atlantic Ocean". Is this according to fig. 2 it is +/- 5°?

Here we modified to 'covering ~8-70 Ma old lithosphere between 4°N and 4°S in the equatorial Atlantic Ocean'. The equatorial Atlantic oceanic ridge can be defined as the ridge segments between the Vema TF at ~10°N and the Ascension TF at ~7°S [Udintsev, 1996], but we didn't specify this definition in the manuscript because it is not related to the main idea of this work.

Line 118: Make clear that these are converted (PS) or since hydrophones were used for picking PSP converted. What is the conversion horizon? For young crust with no sediments most often it is the seafloor itself.

In the main text we clarified that the picked S-wave arrivals have P-to-S and S-to-P conversion modes at the interface between the igneous crust and the sedimentary layer (if present) or water. And we provided detailed analyses in Section 'Identifying the P-to-S and S-to-P conversion interfaces' in Methods and Supplementary Figs. 3-5.

Line 134: Does tectonic accretion need a detachment fault? I understood that detachments develop under certain conditions regarding the magmatic budget. Too less or too much magma, no detachment development. Compare to Buck et al., 2005. $M = \text{Magmatic Budget}$; $M \leq 0.5$

Here we deleted 'along detachment fault'.

Line 143: add PmP  "... reflections (PmP)."

Modified as suggested by the reviewer.

Lines 161-163: It looks like you observe a seismic Moho in that segment. What does this reflector stand for? A really seismic Moho separating igneous crust from mantle or a serpentinisation front or any other explanation?

We observed a seismic Moho along the whole profile except in Segment 3-N. We interpret this seismic Moho as the boundary between crust and mantle, which is characterized by rapid increase in both P- and S-wave velocities across the Moho from the crustal to the mantle. The clear P- and S-wave reflections off the Moho demonstrate a relatively thin or sharp boundary between the crust and mantle. Though our tomographic method cannot constrain the thickness of the Moho transition zone, this seismic Moho is not likely represents a serpentinization front as the serpentinization process tends to be gradual leading to more smooth velocity variation in depth.

Line 178: "a slight thinning" <-- quantify slight. Or is this represented by 5.4 km total crustal thickness?

Here we modified to 'The oceanic crust shows a slight thinning in a narrow zone in the vicinity of the FZ, TF and pseudo-fault regions, where the thinnest crust at these tectonic discontinuities is <1.5 km thinner relative to the average of each segment '.

Line 181: Crust thins by ~600 m to a total crustal thickness of xx km? Does your total crustal thickness include sediments or not? In line 174 you state a total change in crustal thickness of 200 m. This contradicts line 181 with ~600 m change in crustal thickness.

At the beginning of Section 'Intra-segment crustal thickness variation within magmatic segments', we clarify 'the thickness of the igneous crust is defined as the thickness between the top basement and the seismically constrained Moho'. So all the crustal thicknesses discussed in this manuscript don't include the sediments.

Here we changed to 'At the southern end of Segment 1, the crust thins by ~600 m to ~4.8 km thick at the centre of the northern St. Paul FZ valley over <5 km distance'.

Lines 227-231: The authors propose three hypotheses to explain the small variation in crustal thickness in old crust (>7 Ma). However, in the following sections the discussion of these three hypotheses concentrates on evidences for hypothesis 1 mainly. Hypothesis 2: Stretching here refers to the segment centre after crustal formation? I see this hardly discussed in the following sections. A stretching of the segment centre only can be ruled out as long as magnetic anomalies are ridge parallel. Hypothesis 3: I don't see this discussed in the following sections. Why do you rule out this 2 stage accretion?

As mentioned above, we have discussed the three hypotheses in the 'Discussions' section more in-depth. We have elaborated on all the three hypotheses, rejected two as detailed in the response to the second question of reviewer and in the discussion section in the main text.

Line 273: Remember that you are not studying the segment during the time of crustal formation (first stage).

Here we changed to 'a 3-D diapiric mantle upwelling would produce a large variation in crustal thickness within a ridge segment [*Lin and Morgan, 1992; Lin et al., 1990*], which is inconsistent with our observations of the five crustal segments studied here.'

Line 276: Did you look at the mantle bouger anomaly at the recent segment centres of this five segments studied in this work?

Marjanović et al. [2020] processed the high-resolution gravity data collocated with our seismic profile collected during the ILAB-SPARC experiment. Overall, the MBA displays short-wavelength artefacts, not long-wavelength variations with low MBA at segment centres and high MBA at segment ends.

To support our interpretation of presence of uniform crust in the study region, we also compared the along-axis variation of seafloor depth and MBA along several sections of the ridge axis that show focused (Lucky Strike and H₂O) and uniform crustal (Equatorial MAR and MAR at 30°S) variations but here we show two extreme examples for the Lucky Strike segment at 37°N and the segment between the St. Paul and Romanche TFs (Supplementary Fig. 14). The Lucky Strike segment shows enhanced and focused magmatic accretion at the centre of the segment [Seher *et al.*, 2010; Singh *et al.*, 2006], leading to large variations in seafloor depth (1.3-2.8 km) and MBA (~30 mGal) from the centre to the distal ends (Supplementary Fig.14 a,b). In contrast, the segment between the St. Paul and Romanche TFs shows <0.7-1.0 km variation in seafloor depth and <15-20 mGal variation in the MBA (Supplementary Fig.14 c,d), much smaller than those observed along the Lucky Strike segment but comparable with variations along the fast-spreading ridges (200-700 m and 10-20 mGal, respectively; [Lin and Morgan, 1992]), supporting the presence of relatively uniform crust.

Lines 295-297: Give a reference for the "thick crust".

Modified as suggested by the reviewer.

Lines 353-364: This seems to be a kind of doubling to the lines 328-331.

In Line 328-331, we discussed the relation between the along-axis crustal thickness variation within a second-order ridge segment and the length of the second-order ridge segment. In Line 353-364, we discussed the relation between the along-axis crustal variation and the length of the adjacent oceanic TF. So it's not doubling. However, we have removed the discussion of the relationship between crustal thickness variation and the length of second-order ridge segment in the current manuscript.

Line 384: 50% versus 90% magmatic accretion < You look at very different times of crustal accretion within the 5 segments. You might image magmatically active times of the specific segments.

In this work, we study the thickness of crust within five crustal segments formed 8, 24, 40 and 70 Ma, respectively. We don't believe it's a coincidence that we sampled five segment formed during magmatically active times, though we have no strong evidence to rule out this possibility. However, all the five ridge segments seem to be magmatically robust, which is supported by the uniform magmatic crust (Fig. 5) and the well-defined Moho boundary (Figs. 3 and 4a).

Line 394: Why did you change frequency, was the signal for Pn so much improved but Pg and PmP haven't been visible any longer?

Filtering the seismic data from 4-20 Hz is sufficient for picking the near-offset (<100 km) mantle refractions (Pn) but is difficult for picking Pn arrivals at far offset up to 700 km [Wang *et al.*, 2022]. To pick the Pn arrivals up to 700 km offset, we filtered the data to a maximum frequency of 15 Hz. To

keep the consistency of the frequency band, we filtered the data from 4 to 15 Hz for picking Pn arrivals at both near and far offsets.

Lines 457 and followings: You observe two times converted S-Waves. Starting as P-Wave at the source, converting at or in the subsurface, travelling as S through the subsurface becoming converted to P at a boundary or seafloor and than being recorded as P-Wave to the hydrophone. Where does the conversion takes place? If at the seafloor, did you take the sedimentary cover into account? It will extremely delay the S-wave arrivals even if only a few centimeter of sediments.

We clarified in the main text that the picked S-wave arrivals have P-to-S and S-to-P conversions at the interface between igneous crust and the sediment (if present) or water. We have also provided detailed analyses in Section 'Identifying the P-to-S and S-to-P conversion interfaces' in Methods and in Supplementary Figs. 3-5.

Line 467: add km to "... 60 to 760 horizontal distances ..."

Modified as suggested by the reviewer.

Line 468: change to something more clear maybe: "the lower crust has sparse ray coverage, PmP only."

We rewrote to clarify this point.

Lines 476-477: You do not have Vs there at all.

We have clarified this point.

Line 486: "50 models"  "50 calculated models" or "50 inverted models"

Modified as suggested by the reviewer.

Line 512: "...models is 20x3 (Supp..." add "km". Did you also run a checkerboard test with inverted anomalies, to show that the resolution is independent of the polarity of the anomaly?

As suggested by the reviewer, we performed checkerboard tests with inverted velocity anomalies and also with inverted variation in Moho depth (see Methods). The results are shown in Supplementary Figs. 9-11. Our tests suggest the resolution of the used tomography method is independent on the polarity of the anomaly.

Lines 529-531: This is a very weak explanation, because you do not allow for Moho depth adjustments. This does not mean that this Vp-Moho is the best possible. By this you do not allow for more variation.

We deleted this sentence in Lines 529-531.

Fig. 4: Panel b, segment 3N, shows models with a negative vertical velocity gradient which hints to problems in the inversion or how do you explain a negative velocity gradient in your models?

The negative vertical velocity gradient is observed in the 1-D velocity profile because the lower crustal velocity in this region is not constrained by PmP arrivals. The negative vertical velocity gradient is produced by the smoothing effect of tomography. In this work, we only discuss the crustal velocity structure within ~2.2 km sub-basement depth for the segment 3-N, which is constrained by the dense Pg arrivals.

Suppl. Fig. 5: panel d, I do not see the advantage of presenting the variance of V_p/V_s , since V_p is kept constant it more or less reflects the variance of V_s .

We removed the plot of the standard deviation of V_p/V_s .

Reviewer #2:

This manuscript presents a new V_p and V_s model for the oceanic crust across several segments in the equatorial Atlantic. The seismic line spans a large range of crustal ages. The velocity modeling shows that (1) the crust was magmatically accreted, and (2) crustal thickness is roughly constant, in contrast to other slow-spreading crust where crustal thickening is observed toward segment centers. The authors interpret this crustal thickness observation as indicating that the crust in this part of the Atlantic accreted magmatically via 2D sheet-like mantle upwelling rather than the 3D plume-like upwelling generally hypothesized for slow-spreading ridges.

This is an interesting result, and is well presented in the manuscript. It has important implications for how we conceptualize crustal accretion and mantle upwelling at ridges. I do have some questions and comments, which are listed below. The main points that I think need addressing are these: (1) the resolution tests could do a better job of convincing readers that the uniform crustal thickness is a robust result; and (2) the manuscript makes some statements about the implications of the study that seem to equate magmatic upwelling at ridges with 2D sheet-like upwelling, although elsewhere the existence of 3D plume-like upwelling at some slow-spreading segments is discussed. I suspect the first issue is only because of the choice of testing parameters, and the second can be addressed by refining some of those over-broad statements.

We have provided extensive resolution tests as pointed out by the reviewer and have revised the manuscript to address the second point.

I understand the desire not to speculate too much without solid evidence, but I would be interested to read more of the authors' thoughts on why magmatic crust in the equatorial Atlantic is apparently accreting differently than elsewhere along the MAR. Is it the presence of extra-long transforms and their stabilizing effects on ridge thermal structure that make this 2D upwelling possible in a slow-spreading regime? This is suggested in the discussion, but the abstract and final conclusions focus on the potential to map out 2D upwelling regions rather than ideas as to why this upwelling scenario exists. Discussion of the role of these long transforms seems especially relevant in the context of the related manuscript on the velocity structure beneath Romanche.

In the revised manuscript, we proposed two mechanisms that likely could facilitate a 2-D mantle upwelling beneath slow-spreading ridges in the equatorial Atlantic Ocean: (1) *en échelon* large oceanic transform faults (TFs) and (2) higher CO₂ and H₂O concentrations in the mantle melt in the equatorial Atlantic region. Please refer to the response of the first question from reviewer 1 and the discussion section in main text for the details.

Specific comments:

L18: I am not sure that “contradicting [a] previous hypothesis” is accurate here. I assume that the “previous hypothesis” is supposed to be that 2D sheet-like upwelling cannot occur at slow-spreading ridges, though this isn't explicitly stated. Either that previous hypothesis should be more clearly defined (if it is actually a previous hypothesis and not a straw man), or I would suggest rephrasing this to state that “...uniform magmatic accretion at slow spreading rates is due to a two-dimensional sheet-like mantle upwelling more similar to magmatism at fast-spreading ridges than at previously surveyed slow-spreading ridge segments.”

We thank reviewer for the above remark and suggestion. We have modified the text accordingly.

L212-214: This is slightly concerning – if this inversion could be missing as much as 1 km of crustal thickness variation, then it is difficult to argue that the crustal thickness here is unambiguously uniform (and, in particular, more uniform from ends to centers than other MAR crust – if your points on Fig 1 moved up 1 km on the vertical axis, they would start to cluster with the other MAR points). The final sentence of this paragraph is “These tests demonstrate that the observed uniform crust within these five ridge segments is real” (L215). Please clarify the justification for that last sentence.

Here we performed the checkerboard tests using checkerboard pattern with a velocity anomaly of 10% and Moho depth perturbation of different half-wavelengths (50, 100 and 200 km, respectively). In

these tests, the variations in Moho depth in the checkerboard models are ~2.5-3.0 km. The final inverted results (Supplementary Figs. 9-11) show that the used travel time tomography method can recover the Moho depth and its lateral variation for most portions along our seismic profile. So if large crustal thickness variations exist within the five crustal segments we have studied here, we should observe large crustal thickness variations in the tomographic model. However, the tomographic crustal structure shown relatively uniform thickness, which in turn, demonstrates no large crustal thickness variations within the five segments.

Fig 6 and 7, L514-515: Related to the previous point, why was the crustal thickness perturbation wavelength for the checkerboard tests set at 70 km? Given that the recovered crustal thickness from that perturbation is significantly more uniform than the input model, it's not a very convincing argument for this model being able to properly resolve crustal thickness, but perhaps a longer perturbation wavelength could be better resolved. A longer wavelength would also be relevant to the question of whether the crustal thickness is uniform on a segment scale.

As suggested by the reviewer, we performed the checkerboard tests using models with variation of different lateral half-wavelengths in Moho depth (50, 100 and 200 km). The inverted results are shown in Supplementary Figs. 9-11. These tests demonstrate that the used travel time tomography method can almost recover the Moho depth and its lateral variation for most portions along our seismic profile.

L275 paragraph: It seems like there are two arguments here mixed together: (1) uniform crustal thickness implies uniform (non-3D) upwelling, and (2) there are long-wavelength trends in MORB geochemistry that aren't segment-scale and don't correlate to transform faults, indicating that the presence of transforms isn't controlling mantle temperatures enough to make upwelling more 3D and diapiric. I think the geochemistry part should be a paragraph on its own. The first part is more of a restatement of the main argument of the paper.

We rephrased this paragraph as suggested by reviewer. In the revised manuscript, we first concluded that the relatively uniform crustal thickness observed in the equatorial Atlantic Ocean is due to a relatively uniform (nearly 2-D) mantle upwelling beneath the ridges at the time of crustal accretion. And in the following paragraph, we used the results of long-wavelength trends from geochemistry as an evidence to support our conclusion of a nearly 2-D mantle upwelling in the study region.

L338 paragraph: This paragraph also seems to contain more than one big idea: (1) some kind of balance between TF enhanced melting and hydrothermal cooling at segment centers might lead to relatively flat isotherms and uniform upwelling (up to L353), and (2) the length of a transform (but not specifically the offset on the transform) might have some effects on thermal structure, with mega-transforms facilitating this 2D upwelling scenario. I would recommend separating the first part of the

paragraph from the second part, and reordering the paragraph for the second part to put the main point upfront.

We thank the reviewer for the suggestion. We have now separated these two ideas. The balance between transform fault (TF) enhanced melting and hydrothermal cooling at segment centres could lead to a relatively flat isotherm. And we used this to argue that the melt is not necessarily focused to the segment centres at the base of the lithosphere as suggested by *Magde and Sparks* [1997] and some later works. This is further supported by the geochemistry study which suggests that the melt does not focus to a magma chamber but erupts rapidly vertically.

In the revised manuscript, we propose that the *en échelon* large oceanic TFs facilitate the uniform mantle upwelling in the equatorial Atlantic region, using a compilation of segment length and transform offset to support this idea.

L386: I am not sure what “suggesting that the mode of accretion plays a more important role in defining the 2-D versus 3-D mantle upwelling” means here. Does “mode of accretion” refer to tectonic vs magmatic accretion? If this is intended to imply a connection between the (large) percentage of magmatically accreted crust in the equatorial Atlantic and the mode of upwelling, i.e. 2D upwelling at slow spreading rates is possible if most of the crust is magmatically accreted, then I’d like to see more evidence for that connection/discussion of some mechanism. Otherwise it sounds kind of like correlation implying causation.

We have deleted this sentence to avoid confusion.

L387-389 (and also L19-20): I am not convinced by this idea that the “lateral extent of magmatic accretion could be used to map the 2D sheet-like mantle upwelling regions along the global spreading ridge system.” Maybe what is meant is that the lateral extent of uniform-thickness magmatic-style crust can be used to define where sheet-like upwelling has occurred? “Magmatic accretion” also encompasses crust with segment-scale thickness variations that was accreted via 3D plume-like magmatic processes at slow-spreading ridges.

We have deleted this sentence as this idea is not well developed and is not really helpful to the main conclusion of this paper.

Some picky comments on grammar/typos/figures:

L18: “contradicting previous hypotheses” or “contradicting the previous hypothesis” - as written, the singular “hypothesis” doesn’t work.

Modified as suggested by the reviewer.

L19: the lateral extent

Modified as suggested by the reviewer.

L284-287: For the sentence that starts “While north of the St. Paul fracture zone...” I think if you remove the word “While” it would make sense grammatically.

Modified as suggested by the reviewer.

L403: 50 or 70 ms? Do you mean 50-70 ms?

The picking uncertainty of the PmP arrivals is 50 ms or 70 ms; we have clarified this point.

Supp. Fig 2: Why shift times by 2 seconds rather than having the time axis start at 2? It’s not incorrect or anything, but I’m genuinely curious if there’s a reason for doing this.

We shifted the travel time by 2 s to simultaneously show the P-wave first arrivals and the S-wave arrivals in the same plot, otherwise the S-wave first arrivals with offset >10 km would appear at negative time when the reduction velocity is 4 km/s. So the 2 s shift is only for display purpose. We have clarified this point in the figure caption.

L467-468: for the 60 to 760 “horizontal distances” is the unit km along the seismic line?

Modified as suggested by the reviewer.

Supp. Fig 6c and 7c: Could you use more distinct colors than red and magenta for the two Moho lines? I can’t tell them apart.

We have modified the figures to address the above point.

Fig 1: How is the boundary between uniform and segment center-focused crust defined (dashed black horizontal line)? Is it just set at the smallest known thickness variation for non-equatorial Atlantic segments? Please add a citation or provide justification (or remove the line – not sure what purpose it serves since the groups of data points are already fairly distinct).

We have removed the bold dashed line showing 2.8 km crustal thickness difference from Fig. 1.

Fig 2: It would be useful to have a figure panel showing the (half-)spreading rate at the time of formation along this seismic line, so we don’t have to try and match fig 2b to the mapped isochrons in fig 2a. This could replace 2b, or be an additional panel.

As suggested by the reviewer, we have plotted the half-spreading rate of the ridge at the time the studied five segments were formed (see Fig. 2b in main text).

Fig 3, 5, etc: Why is there a gap between segments 2 and 3N? It seems odd particularly because the gap is narrower than the unconstrained part of the Vs model in the Romanche FZ, at least in the bars across the top of Fig 3a.

In our discussion, the Segment 2 extends from the centre of the Romanche transform valley to the centre of the northern valley of the Saint Paul fracture zone. For Segment 3-N, we could define it from ~340 km horizontal distance to the centre of the Romanche transform valley. *Gregory et al.* [2021] argued that the crust below the Romanche transform valley is primarily composed of mafic rocks. In this case, we defined the northern end of the Segment 3-N at the southern bound wall of the Romanche transform valley. This doesn't influence our interpretations.

Figs 1 and 7: caption for 1 says only active source-derived crustal thicknesses are used, but the caption for 7 says data come from both seismic and gravity data; the number of points is the same for the two figures. Are any of them from gravity data?

The maximum crustal thickness variation between the Pico Offset-Oceanographer TFs [*Detrick et al.*, 1995] and the Atlantis TF-Kane TFs [*Lin et al.*, 1990] are obtained from gravity data. All the other estimates are from active-source seismic studies. We clarified this point in Supplementary Table 3.

Reviewer #3:

What are the noteworthy results? Noteworthy is that the authors find relatively uniform crustal thickness at 5 off-axis ridge segments in the central Atlantic. In addition, 4.5 out of 5 segments appear constructed magmatically indicating two-dimensional mantle upwelling and crustal accretion.

Will the work be of significance to the field and related fields? How does it compare to the established literature? If the work is not original, please provide relevant references.

This result is surprising because previous results in both the northern and southern Atlantic Ocean show: i. Frequent amagmatic crustal accretion at the MAR especially near major first and second order offsets. ii. Significant observed variations in crustal thickness within MAR segments.

Existing crustal thickness variations have motivated two models in the established literature: (i) three-dimensional plume-like mantle upwelling; and (ii) two-dimensional mantle upwelling with melt focusing to segment centers along the topography at the base of the lithosphere.

- The Discussion argues against both of these two models for the equatorial Atlantic.
- I am under the impression that the Discussion also argues against these models more broadly in the paragraph comparing crustal thickness variations to 1st and 2nd order segment lengths (lines 315-336) for a compilation of slow spreading ridges. However, it is not quite clear what process this comparison is seeking to test. Clarify the purpose of the paragraph making this comparison.

We have rephrased this paragraph to focus on discussing whether the patterns of mantle upwelling and crustal accretion at slow-spreading ridges are related to the first-order ridge segment length. *Christeson et al.* [2020] proposed that a longer first-order ridge segment can facilitate larger mantle upwelling. Our compilation (Fig. 7) demonstrates that the length of first-order ridge segment has no influence on the crustal accretion process, hence on the mantle upwelling pattern.

Does the work support the conclusions and claims, or is additional evidence needed? This work is lacking in that a new model that explains both these new results and the existing observations is not clearly given in the abstract or the conclusions. Of notable relevance to developing a new model are the facts that: 1. this region of the mid-ocean ridge is broken up by very long offset transform faults and is a noted mantle cold spot (though this is confusing with additional discussion of a Cobb hotspot). This is not mentioned in the abstract. 2. the equatorial Atlantic is a cold spot and this appears crucial to the final interpretation in the very last (concluding?) paragraph (line 374-375, line 378) .But adding this at the very end of the manuscript seems like a suddenly introduced new hypothesis. Specifically, this fact was not mentioned in the abstract and introduction. 3. existing observations seems to support more drastic crustal thickness variations along active ridge segments than off axis. This is mentioned and models related to this observation are rejected, however, an explanation for this observation is not provided.

In the revised manuscript, we proposed two mechanisms that could facilitate a 2-D mantle upwelling beneath slow-spreading ridges in the equatorial Atlantic Ocean: (1) *en échelon* large oceanic transform faults (TFs) and (2) higher CO₂ and H₂O concentrations in the mantle melt. And we also highlight this in the abstract. Please refer to the response to the first question from reviewer 1 and the discussion section in main text for the details.

In the Discussion, paragraph starting at line 275, a convincing argument is made that in the equatorial Atlantic the new crustal thickness measurements and existing geochemical observations support a 2D style of mantle upwelling and crustal accretion.

The model that the authors appear to favor is buried in the Discussion (lines 362-364): “this suggests the mega-transform could facilitate a stable 2-D sheet-like mantle upwelling and a relatively uniform crustal accretion”. This seems to be the authors main conclusion and needs to be included in the

Abstract. An outstanding question that I have about this process is whether the relatively cold mantle temperature in the region is adding to the effect of mega-transforms in controlling the mantle thermal structure.

In the revised manuscript, we proposed two mechanisms that could facilitate a 2-D mantle upwelling beneath slow-spreading ridges in the equatorial Atlantic Ocean: (1) large oceanic transform faults and (2) higher CO₂ and H₂O concentrations in the melt. And we highlight this in the abstract. From our observations, we cannot conclude whether the relatively cold mantle temperature in the region is adding to the effect of mega-transforms in controlling the mantle thermal structure.

Instead what the structure of the paper conveys to be more important is the poorly developed idea that: “the lateral extent of magmatic accretion could be used to define the sheet-like mantle upwelling regions beneath the global ridge system”. It is simply stated at the end of the Abstract (note that when reading it I could not understand what “lateral extent” referred to). The next mention is in the very last sentence of the manuscript: “and the lateral extent of the magmatic accretion could be used to map the 2-D sheet-like mantle upwelling regions along the global spreading ridge system”. Why is this important? This point does not seem as relevant to me as featuring the main interpretation/model in the Abstract.

Here we deleted this sentence.

The final paragraph (lines 386-387) also states: “that the mode of accretion plays an important role in defining the 2-D versus 3-D mantle upwelling”. What exactly is the thinking here? Does the mode of accretion control whether mantle upwelling is 2-D versus 3-D, i.e. is the control top-down? Or is the mode accretion a consequence of the pattern of mantle upwelling, i.e. the control is bottom-up? I believe the second is being argued here – using language that is more clear than “defining” might help clarify the intent of this statement.

Here we deleted this sentence.

Are there any flaws in the data analysis, interpretation and conclusions? Do these prohibit publication or require revision?

- The seismic data shown in the supplement looks good. There are very nice S wave arrivals – a consequence of collecting this data off-axis where acoustic waves can convert into S waves in the seafloor sediments at the ray entry point.
- Lacking is a specific test of whether a model with along axis variations in crustal thickness is precluded by the data. This is required to support the interpretation that crustal thickness is more or less uniform along the 5 ridge segments. Specifically, the variable ability to recover the Moho

checkerboard pattern raises this question (lines 212-214 & Supplemental Figures 6-8). A discussion of the relative weight of travel times for direct P phases versus those for reflected phases on the results is not included.

Firstly, we performed the checkerboard tests using checkerboard pattern with size of 20×2 km and 10% velocity anomaly. And we added Moho depth perturbation of different half-wavelengths (50, 100 and 200 km) in the checkerboard model. In these tests, the variations in Moho depth in the checkerboard models are ~ 2.5 -3.0 km. The final inverted results (Supplementary Figs. 9-11) show that the used travel time tomography method can recover the Moho depth and its lateral variation for most portions along our seismic profile. And the resolvability of the used tomography method is independent on the polarities of the velocity anomaly and the Moho depth perturbation. This means if large along-axis crustal thickness variations do exist, our traveltimes method can recover it.

Secondly, the results from the Monte-Carlo analysis [Korenaga *et al.*, 2000] by taking the variations of regularization parameters into account show similar crustal V_p structure with relatively uniform crust, which suggests the inversion of the picked P_g and P_mP travel times is robust. The standard deviation of the Moho depth is < 400 m. This further supports that the uniform crust along our seismic profile is real. Here we presented two inverted results which obtained using parameters allowing large updating in Moho depth (see Fig. A2 below). For these two models, the Moho boundary shows more variability, but the crust still shows little variations in thickness, where the standard deviation of average crustal thickness is < 0.4 km.

Figure A2: (a,b) Final inverted results obtained using parameters allowing large updating in Moho depth. (c) and (d) show the crustal thickness variations (in black) and average crustal thickness (in red) of each segment for final inverted model (a) and (b), respectively. The values in red in plots (c,d) represent the average crustal thickness and the standard deviations.

- The effect of varying the regularization parameters on the velocity structure and Moho topography is not explored in the Monte Carlo analysis.

As suggested by the reviewer, we performed the Monte Carlo analysis for crustal P-wave velocity structure by considering the variation of the regularization parameters on the velocity and Moho structure. (see Section ‘Monte-Carlo analysis’ in Methods). The results are shown in Supplementary Fig. 8. The variance in the final crustal V_p model is less than 0.1 km/s in the upper crust and is less than 0.3 km/s in the lower crust (Supplementary Fig. 8a). The maximum standard deviation of the Moho depth is ~ 400 m (Supplementary Fig. 8b). The preferred Moho (Fig. 3a) falls in the standard deviation of the average Moho depth from the Monte-Carlo analysis (see red curve in Supplementary Fig. 8b). Similar Monte-Carlo analyses are performed to assess the variance in the crustal V_s . Supplementary Fig. 8c shows the variance of the crustal V_s calculated using 50 final inverted models.

For most portion of the crust, the crustal Vs has a variance <0.1 km/s, and large variances >0.1 km/s are observed around the TF and FZs and at the southern and northern extremity of the model.

- To support the suggestion that colder mantle is playing a role in generating an average crustal thickness of 5.5 km (lines 379-381) the paper needs to include a calculation that shows whether or not a 150 °C reduction in mantle is consistent with a reduction in crustal thickness/melt production of 500 m.

Schilling et al. [1995] estimated that the minimum melting temperature is ~1300 °C in the equatorial Atlantic Ocean. The numerical modelling of *Behn and Grove* [2015] demonstrated that a ~5 km thick crust can be formed when the half-spreading rate is ~20 mm/year and the mantle potential temperatures is 1300 °C, which is 500 m thinner than the estimates (~5.5 km) we obtained in this paper. The large amount of volatiles (CO₂ and H₂O) in the mantle in the equatorial Atlantic Ocean will decrease the mantle solidus and increase the depth extent of the melting regime, leading to the enhanced production of melt. The enhanced melt supply could increase the crustal thickness. In the revised manuscript, we delete that convincing expression of 150 °C temperature reduction in the equatorial Atlantic Ocean.

Is the methodology sound? Does the work meet the expected standards in your field?

- The seismic analysis seems sound and uses a well-established method

Is there enough detail provided in the methods for the work to be reproduced?

- Yes. However, the choice of regularization parameters for the preferred model are not given.

We have provided more details of the traveltimes tomography in the revised manuscript. In our traveltimes tomography, both first- and second-order velocity regularizations are imposed to obtain a smooth velocity model [*Van Avendonk et al.*, 2004]. The weight given to the horizontal derivatives is 4 times of that given to the vertical derivative, following *Van Avendonk et al.* [2004] and *Roland et al.* [2012]. The regularization parameters are tested and selected in each iteration step to avoid the introduction of artefacts. We use the standard χ^2 value [*Van Avendonk et al.*, 2004] to measure the mismatch between the modelled and manually picked travel times. Large regularization values are used at the early stage of tomography and the regularization values are reduced when χ^2 value approaches 1. These points have been added in the 'Methods' section.

Detailed comments:

- Abstract: The two existing models for crustal thickness variations at slow spreading ridges are incorrectly merged together in the phrase (lines 11-12) "due a three-dimensional plume-like mantle upwelling with melt focusing to segment centres".

We have modified this sentence.

- Introduction – issues with the description of crustal accretion at fast spreading ridges:
 - o Line 32:- Delete NTOs: Second order offsets are OSCs at fast spreading ridges and NTOs at slow spreading ridges

Modified as suggested by the reviewer.

- This statement ignores the literature that crust is not thin beneath OSCs at the EPR (e.g. Canales et al., 2003).

Modified as suggested by the reviewer.

- o Lines 33-34: Incorrect: fast spreading ridges are fed by individual mantle upwellings (e.g. Toomey 2007) and are not just two-dimensional sheet-like mantle upwellings.

At fast-spreading ridge, the relatively uniform crust is generally interpreted as due to a uniform, 2-D, sheet-like mantle upwelling beneath ridge axis [*Lin and Morgan, 1992*]. Here the sheet-like mantle upwelling refers to the mantle upwelling within a second-order ridge segments, not the first-order ridge segments between two major transform faults. We agree that the overall mantle upwelling between two major transform faults is not perfectly sheet-like due to the offset of overlapping spreading centres. We went through the paper from *Toomey et al. [2007]* in *Nature*. They proposed the mantle upwelling beneath the fast-spreading East-Pacific Rise could be skewed. But *Singh and Macdonald [2009]* pointed out that the results from *Toomey et al. [2007]* are not reliable.

- Supplemental Figures 6 &7: Caption reads that this is for Vp model, figures are labelled as being S models

Modified as suggested by the reviewer. In these figures, ‘S’ means south direction. We replotted the figures to avoid confusions.

In summary, I recommend major revisions so that the manuscript more clearly argues for the main processes that the authors infer from these new observations.

References

- Behn, M. D., M. S. Boettcher, and G. Hirth (2007), Thermal structure of oceanic transform faults, *Geology*, *35*(4), 307-310.
- Behn, M. D., and T. L. Grove (2015), Melting systematics in mid-ocean ridge basalts: Application of a plagioclase-spinel melting model to global variations in major element chemistry and crustal thickness, *J. Geophys. Res.*, *120*(7), 4863-4886.
- Cann, J. R., et al. (1997), Corrugated slip surfaces formed at ridge–transform intersections on the Mid-Atlantic Ridge, *Nature*, *385*(6614), 329-332.
- Carlson, R. L., and D. J. Miller (1997), A new assessment of the abundance of serpentinite in the oceanic crust, *Geophys. Res. Lett.*, *24*(4), 457-460.
- Christeson, G. L., et al. (2020), South Atlantic Transect: Variations in Oceanic Crustal Structure at 31°S, *Geochem. Geophys. Geosyst.*, *21*(7), e2020GC009017.
- Combiér, V., et al. (2015), Three-dimensional geometry of axial magma chamber roof and faults at Lucky Strike volcano on the Mid-Atlantic Ridge, *J. Geophys. Res.*, *120*(8), 5379-5400.
- Detrick, R. S., H. D. Needham, and V. Renard (1995), Gravity anomalies and crustal thickness variations along the Mid-Atlantic Ridge between 33°N and 40°N, *J. Geophys. Res.*, *100*(B3), 3767-3787.
- Escartín, J., et al. (1999), Quantifying tectonic strain and magmatic accretion at a slow spreading ridge segment, Mid-Atlantic Ridge, 29°N, *J. Geophys. Res.*, *104*(B5), 10421-10437.
- Escartín, J., and J. Lin (1995), Ridge offsets, normal faulting, and gravity anomalies of slow spreading ridges, *J. Geophys. Res.*, *100*(B4), 6163-6177.
- Escartín, J., et al. (2008), Central role of detachment faults in accretion of slow-spreading oceanic lithosphere, *Nature*, *455*(7214), 790-794.
- Gregory, E. P. M., S. C. Singh, M. Marjanović, and Z. Wang (2021), Serpentinized peridotite versus thick mafic crust at the Romanche oceanic transform fault, *Geology*, *49*(9), 1132-1136.
- Grevemeyer, I., et al. (2021), Extensional tectonics and two-stage crustal accretion at oceanic transform faults, *Nature*, *591*(7850), 402-407.
- Harmon, N., et al. (2018), Marine Geophysical Investigation of the Chain Fracture Zone in the Equatorial Atlantic From the PI-LAB Experiment, *J. Geophys. Res.*, *123*(12), 11016-11030.
- Keller, T., and R. F. Katz (2016), The Role of Volatiles in Reactive Melt Transport in the Asthenosphere, *J. Petrol.*, *57*(6), 1073-1108.
- Korenaga, J., et al. (2000), Crustal structure of the southeast Greenland margin from joint refraction and reflection seismic tomography, *J. Geophys. Res.*, *105*(B9), 21591-21614.
- Le Voyer, M., et al. (2019), Carbon Fluxes and Primary Magma CO₂ Contents Along the Global Mid-Ocean Ridge System, *Geochem. Geophys. Geosyst.*, *20*(3), 1387-1424.
- Lin, J., and J. P. Morgan (1992), The spreading rate dependence of three-dimensional mid-ocean ridge gravity structure, *Geophys. Res. Lett.*, *19*(1), 13-16.

- Lin, J., et al. (1990), Evidence from gravity data for focused magmatic accretion along the Mid-Atlantic Ridge, *Nature*, 344(6267), 627-632.
- Magde, L. S., and D. W. Sparks (1997), Three-dimensional mantle upwelling, melt generation, and melt migration beneath segment slow spreading ridges, *J. Geophys. Res.*, 102(B9), 20571-20583.
- Marjanović, M., et al. (2020), Seismic Crustal Structure and Morpho-tectonic Features Associated with the Chain Fracture Zone and their Role in the Evolution of the Equatorial Atlantic Region, *J. Geophys. Res.*, 125, e2020JB020275.
- Roland, E., D. Lizarralde, J. J. McGuire, and J. A. Collins (2012), Seismic velocity constraints on the material properties that control earthquake behavior at the Quebrada-Discovery-Gofar transform faults, East Pacific Rise, *J. Geophys. Res.*, 117(B11), B11102.
- Schilling, J.-G., et al. (1995), Thermal structure of the mantle beneath the equatorial Mid-Atlantic Ridge: Inferences from the spatial variation of dredged basalt glass compositions, *J. Geophys. Res.*, 100(B6), 10057-10076.
- Searle, R. (2013), *Mid-Ocean Ridges*, Cambridge University Press, Cambridge.
- Seher, T., et al. (2010), Crustal velocity structure of the Lucky Strike segment of the Mid-Atlantic Ridge at 37°N from seismic refraction measurements, *J. Geophys. Res.*, 115(B3), B03103.
- Shaw, P. R. (1992), Ridge segmentation, faulting and crustal thickness in the Atlantic Ocean, *Nature*, 358(6386), 490-493.
- Shaw, P. R., and J. Lin (1993), Causes and consequences of variations in faulting style at the Mid-Atlantic Ridge, *J. Geophys. Res.*, 98(B12), 21839-21851.
- Singh, S. C., et al. (2006), Discovery of a magma chamber and faults beneath a Mid-Atlantic Ridge hydrothermal field, *Nature*, 442(7106), 1029-1032.
- Singh, S. C., and K. C. Macdonald (2009), Mantle skewness and ridge segmentation, *Nature*, 458(7241), E11-E12.
- Toomey, D. R., et al. (2007), Skew of mantle upwelling beneath the East Pacific Rise governs segmentation, *Nature*, 446(7134), 409-414.
- Udintsev, G. (1996), Equatorial segment of the mid-Atlantic ridge, *Technical Series. Intergovernmental Oceanographic Commission= Série technique*.
- Van Avendonk, H. J. A., D. J. Shillington, W. S. Holbrook, and M. J. Hornbach (2004), Inferring crustal structure in the Aleutian island arc from a sparse wide-angle seismic data set, *Geochem. Geophys. Geosyst.*, 5(8), Q08008.
- Wang, Z., et al. (2022), Deep hydration and lithospheric thinning at oceanic transform plate boundaries, *Nature Geosci.*, 15, 741-746.
- Whitehead, J. A., H. J. B. Dick, and H. Schouten (1984), A mechanism for magmatic accretion under spreading centres, *Nature*, 312(5990), 146-148.
- Wolfson-Schwehr, M., and M. S. Boettcher (2019), Chapter 2 - Global Characteristics of Oceanic Transform Fault Structure and Seismicity, in *Transform Plate Boundaries and Fracture Zones*, edited by J. C. Duarte, pp. 21-59, Elsevier, doi:<https://doi.org/10.1016/B978-0-12-812064-4.00002-5>.

REVIEWERS' COMMENTS

Reviewer #1 (Remarks to the Author):

The authors reacted on every single remark that reviewers gave for the manuscript on uniform crustal accretion along slow-spreading ridges in the equatorial Atlantic Ocean.

In my opinion unclear points have been explained, changed or taken away. Changes to text and figures have been made and I would recommend the manuscript for publication in its current state from the point of science.

Best regards

Reviewer #2 (Remarks to the Author):

The authors have addressed my initial review thoroughly, and I appreciate the time and effort they put into revisions. I think the revised discussion is much clearer, and shifting the focus from mapping out different types of crustal accretion to more consideration of the mechanisms that might cause this distinctive mode of magmatic accretion in the equatorial Atlantic made the paper more interesting to me, at least.

I have one tiny comment on wording: line 291 of the revised manuscript says that the crust is "not thick enough, and hence not hot enough" for ductile lower crust to be a factor - I would guess the intended meaning is that thin (magmatic) crust indicates there wasn't excessive magma production, so the mantle temperature wasn't anomalously high and the lower crust is unlikely to have been unusually hot, but the phrasing is not entirely clear.

Other than that picky note on phrasing, I think the manuscript is in good shape and suitable for publication. The work is significant, the observations support the conclusions, and the discussion is thoughtful.

Reviewer #3 (Remarks to the Author):

The revised manuscript has been substantially improved to address the reviewers' comments. In particular, the interpretation and of the results and reasoning much more clearly explains the significance of the results: that the equatorial Atlantic is different from the north and south Atlantic likely due to the effect of long transform faults on the mantle thermal structure and/or the presence of high water and CO₂ in the mantle source.

Two minor revisions remain to be made:

1. The last sentence of the manuscript (lines 424-425) concludes that 2D upwelling is the norm at slow spreading ridges. However, this is not what the paper argues. Instead the paper argues that there is 2D upwelling in the equatorial Atlantic and while there is significant along axis variation in crustal thickness in the northern and southern Atlantic. Rephrase to be consistent with the reasoning in the text. The same for the text on lines 363-366.
2. Paragraph on tectonic extension (lines 247-256): The argument here is focused on tectonic modification of the segment ends. I am of the same opinion as Reviewer 1 that the option of tectonic modification and crustal thinning of the segment centers should also be discussed in order to fully address the proposed hypotheses.
3. I am attaching a marked up PDF with many small English grammar edits for the main body of the text and the figure captions.

Reviewers' comments are shown in black; authors' response is shown in blue.

Reviewer #2:

The authors have addressed my initial review thoroughly, and I appreciate the time and effort they put into revisions. I think the revised discussion is much clearer, and shifting the focus from mapping out different types of crustal accretion to more consideration of the mechanisms that might cause this distinctive mode of magmatic accretion in the equatorial Atlantic made the paper more interesting to me, at least.

I have one tiny comment on wording: line 291 of the revised manuscript says that the crust is "not thick enough, and hence not hot enough" for ductile lower crust to be a factor - I would guess the intended meaning is that thin (magmatic) crust indicates there wasn't excessive magma production, so the mantle temperature wasn't anomalously high and the lower crust is unlikely to have been unusually hot, but the phrasing is not entirely clear. Other than that picky note on phrasing, I think the manuscript is in good shape and suitable for publication. The work is significant, the observations support the conclusions, and the discussion is thoughtful.

We have rephrased this sentence to 'the crust formed at the MAR in the equatorial Atlantic Ocean is much thinner (5.5-5.6 km) than that formed at the Reykjanes Ridge, indicating that the mantle is colder, and therefore the lower crust is not hot enough to enable rapid ductile flow within the lower crust.'

Reviewer #3:

The revised manuscript has been substantially improved to address the reviewers' comments. In particular, the interpretation and of the results and reasoning much more clearly explains the significance of the results: that the equatorial Atlantic is different from the north and south Atlantic likely due to the effect of long transform faults on the mantle thermal structure and/or the presence of high water and CO₂ in the mantle source. Two minor revisions remain to be made:

1. The last sentence of the manuscript (lines 424-425) concludes that 2D upwelling is the norm at slow spreading ridges. However, this is not what the paper argues. Instead the paper argues that there is 2D upwelling in the equatorial Atlantic and while there is significant along axis variation in crustal thickness in the northern and southern Atlantic. Rephrase to be consistent with the reasoning in the text. The same for the text on lines 363-366.

We have deleted the sentence on lines 424-425 and have rephrased the texts on lines 363-366.

2. Paragraph on tectonic extension (lines 247-256): The argument here is focused on tectonic modification of the segment ends. I am of the same opinion as Reviewer 1 that the option of tectonic modification and crustal thinning of the segment centers should also be discussed in order to fully address the proposed hypotheses.

In this paragraph, we discussed the difference in the amounts of tectonic extension and stretching between segment centres and segment ends by comparing the spacing, heave and throw of normal faults. The spacing, heave and throw of normal faults are generally larger at segment ends than at segment centres, indicating that more tectonic extension occurs at segment ends [Shaw, 1992; Shaw and Lin, 1993]. This conclusion can be made without detailing the features of tectonic faulting at the segment centre and ends. Since the tectonic extension thins the oceanic crust [Combiere et al., 2015; Escartín et al., 1999; Escartín and Lin, 1995], more tectonic extension means more thinning of oceanic crust would occur at the segment ends. So, the overall influence of tectonic extension on crustal thickness variation would enhance the along-axis crustal thickness variation, contrary to our observation of the uniform crust along the profile.

3. I am attaching a marked up PDF with many small English grammar edits for the main body of the text and the figure captions.

We deeply appreciate reviewer for the careful check of English grammar. We have modified accordingly.

References

Combiere, V., et al. (2015), Three-dimensional geometry of axial magma chamber roof and faults at Lucky Strike volcano on the Mid-Atlantic Ridge, *J. Geophys. Res.*, *120*(8), 5379-5400.

Escartín, J., et al. (1999), Quantifying tectonic strain and magmatic accretion at a slow spreading ridge segment, Mid-Atlantic Ridge, 29°N, *J. Geophys. Res.*, *104*(B5), 10421-10437.

Escartín, J., and J. Lin (1995), Ridge offsets, normal faulting, and gravity anomalies of slow spreading ridges, *J. Geophys. Res.*, *100*(B4), 6163-6177.

Shaw, P. R. (1992), Ridge segmentation, faulting and crustal thickness in the Atlantic Ocean, *Nature*, *358*(6386), 490-493.

Shaw, P. R., and J. Lin (1993), Causes and consequences of variations in faulting style at the Mid-Atlantic Ridge, *J. Geophys. Res.*, *98*(B12), 21839-21851.

Shaw, P. R. (1992), Ridge segmentation, faulting and crustal thickness in the Atlantic Ocean, *Nature*, *358*(6386), 490-493.

Shaw, P. R., and J. Lin (1993), Causes and consequences of variations in faulting style at the Mid-Atlantic Ridge, *J. Geophys. Res.*, *98*(B12), 21839-21851.